# KaVa: Latent Reasoning via Compressed KV-Cache Distillation

**Anna Kuzina**[*]
Qualcomm AI Research[†]
akuzina@qti.qualcomm.com

**Maciej Pioro**[*‡]
IDEAS NCBR / IPPT PAN
maciej.pioro@gmail.com

**Babak Ehteshami Bejnordi**
Qualcomm AI Research
behtesha@qti.qualcomm.com

## ABSTRACT

Large Language Models (LLMs) excel at multi-step reasoning problems with explicit chain-of-thought (CoT), but verbose traces incur significant computational costs and memory overhead, and often carry redundant, stylistic artifacts. Latent reasoning has emerged as an efficient alternative that internalizes the thought process, but it suffers from a critical lack of supervision, limiting its effectiveness on complex, natural-language reasoning traces. In this work we propose KaVa, the first framework that bridges this gap by distilling knowledge directly from a compressed KV-cache of the teacher into a latent-reasoning student via self-distillation, leveraging the representational flexibility of continuous latent tokens to align stepwise KV trajectories. We show that the abstract, unstructured knowledge within compressed KV-cache, which lacks direct token correspondence, can serve as a rich supervisory signal for a latent reasoning student. Empirically, the approach consistently outperforms strong latent baselines, exhibits markedly smaller degradation from equation-only to natural-language traces, and scales to larger backbones while preserving efficiency. These results establish compressed KV-cache distillation as a scalable supervision signal for latent reasoning, combining the accuracy of CoT-trained teachers with the efficiency and deployability of latent inference.

## 1 INTRODUCTION

Recent advancements in Large Language Models (LLMs) have demonstrated remarkable capabilities in solving complex problems across domains such as mathematics (Zhang et al., 2025), science (Phan et al., 2025), and code generation (Hui et al., 2024). A key driver of this progress is "chain-of-thought" (CoT) training that elicits intermediate steps before the final answer, improving accuracy on long-horizon inference problems (DeepSeek-AI et al., 2025). Yet, explicit CoT often incurs substantial inference cost due to long, verbose traces and the associated key–value (KV) cache growth, making deployment on memory- and compute-constrained devices difficult. Furthermore, CoT traces, especially those distilled from larger models, can inherit and amplify biases or contain plausible-sounding but fallacious logic, limiting their reliability.

Recent studies show that the KV-caches underlying CoT are highly redundant and can be aggressively compressed with little to no loss in accuracy (Cai et al., 2025; Park et al., 2025), indicating that much of CoT's signal resides in compressible structure rather than indispensable text. This observation suggests an alternative supervisory path: if the essential dynamics of reasoning live in the cache, perhaps models can be trained to internalize those dynamics without verbose traces at inference time. However, this compressed KV-cache presents a significant challenge for knowledge distillation. As pruning decisions are often made independently per layer and attention head, the resulting compressed KV vectors lose their direct correspondence to specific input tokens, rendering conventional distillation schemes that match token activations or layer-wise hidden states ill-posed and non-trivial.

---

[*]Equal contribution
[†]Qualcomm AI Research is an initiative of Qualcomm Technologies, Inc.
[‡]Work done during internship at Qualcomm AI Research.

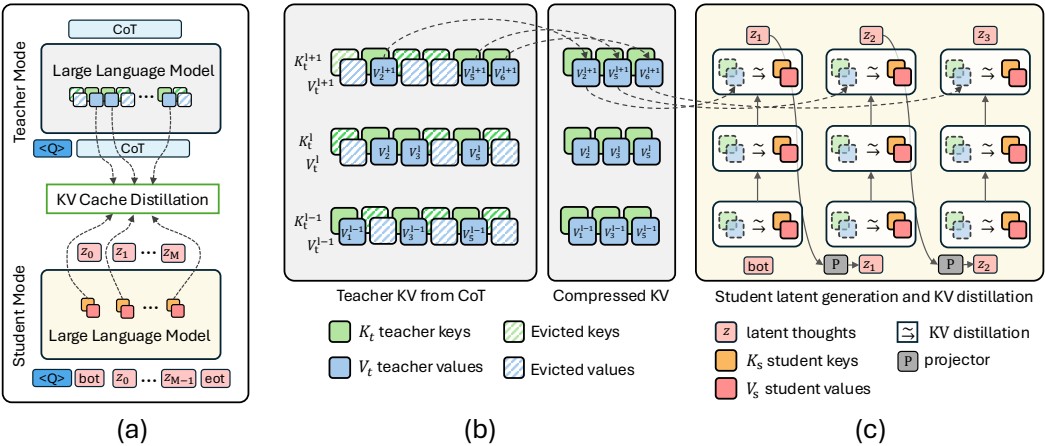

Figure 1: We propose KAVA , a latent reasoning model with KV-cache distillation loss. (a) Overview of our proposed compressed KV-cache distilled latent reasoning framework. (b) Teacher builds full KV-cache from a ground-truth CoT trace; a compression module produces a compressed cache to match the length of the latent trace; (c) a latent-reasoning student generates continuous thoughts $z_t$ and is trained to match compressed teacher KV at each layer/step via KV distillation.

Latent reasoning is a nascent but promising direction in which reasoning occurs within the model's continuous latent space rather than being explicitly externalized (Hao et al., 2024; Su et al., 2025). Latent approaches promise efficiency by reducing token generation and KV-cache footprint, potentially closing the gap between strong reasoning performance and deployability in constrained settings. However, current latent reasoning methods struggle with the absence of direct supervision for internal thoughts, and successes are often reported in restricted setups; performance can degrade when training data contain long, natural-language-style traces that better reflect real-world reasoning workloads. In particular, compared to shorter, template-like traces, models trained on longer, natural-language reasoning sequences exhibit more fragile internal readouts and weaker generalization (Shen et al., 2025; Wu et al., 2025).

In this work, we bridge these gaps by introducing a novel framework that, for the first time, successfully distills the rich, abstract knowledge from a compressed teacher KV-cache into a latent reasoning student. We posit that the continuous, high-dimensional nature of latent thoughts provides a unique representational power that can absorb abstract cache structure that cannot be aligned at the token level. Concretely, our method is composed of three components: (i) the backbone that alternates between a teacher mode that consumes a full CoT to build per-layer, per-head KV-caches and a student mode that generates continuous latent thoughts; (ii) a redundancy- and importance-aware eviction module that compresses the teacher cache to the latent budget; (iii) and a KV-matching loss aligns the student's per-step latent K and V to the compressed target throughout the stack. This yields a strong, stepwise internal supervision signal that teaches the student to "think like" a compact cache of its own explicit reasoning while preserving the inference-time efficiency of latent reasoning. By supervising the latent trajectory directly in KV space, the approach bridges the gap between template-like latent traces and natural-language reasoning, yielding strong gains on natural-language datasets and scaling smoothly to larger backbones while retaining the efficiency benefits of latent inference. Our primary contributions are:

- We are the first to demonstrate that knowledge can be successfully distilled from a compressed KV-cache via self-distillation, despite the cache's head-wise, layer-wise eviction that destroys token correspondence.

- We show that existing KV-cache compression methods can be used to construct a rich, step-by-step supervision signal for latent reasoning. Our method trains a latent student to directly generate this compressed KV-cache at inference time

- Through empirical evaluations, we show that our approach consistently outperforms strong latent baselines on natural language settings, exhibits smaller degradation when moving from equation-only to natural-language traces, and scales to larger backbones.

## 2 BACKGROUND AND RELATED WORKS

**Latent Reasoning.** Traditional reasoning LLMs often rely on generating explicit intermediate steps in language to solve complex reasoning tasks. Recent work shifts reasoning from discrete text tokens to latent continuous tokens, where models perform iterative computation internally without generating external text (Chen et al., 2025; Zhu et al., 2025). Early work validated the benefit of extra computation through unstructured means, such as learnable pause tokens (Goyal et al., 2024) or even semantically meaningless filler tokens (Pfau et al., 2024), which improved performance on reasoning tasks by implicitly extending the model's processing time. Building on this implicit-compute view, iCoT moves from explicit to implicit CoT via distillation (Deng et al., 2023) and curriculum (Deng et al., 2024), progressively removing CoT while aligning internal states around answer prediction. This allows the model to internalize reasoning without generating text rationales at inference. Coconut (Hao et al., 2024) introduces "continuous thought" by feeding the last hidden state directly as the next input embedding, showing breadth-first search–like parallel exploration and fewer thinking tokens versus CoT on logical reasoning tasks. Follow-ups refine supervision and training dynamics: CODI (Shen et al., 2025) compresses CoT into continuous representations via self-distillation that supervises endpoints rather than full trajectories, while PCCoT (Wu et al., 2025) parallelizes latent updates with Jacobi-style iterations to refine multiple continuous thoughts in tandem. In contrast to endpoint- or token-level supervision, our proposed approach distills a teacher's compressed KV-cache into the student's latent trajectory, providing stepwise internal guidance that bridges the supervision gap in continuous-token reasoning without relying on explicit CoT text.

Complementary directions emphasize soft or hybrid traces: SoftCoT (Xu et al., 2025) injects soft thought tokens projected into the backbone's representation space to improve reasoning without altering hard-token generation, and Token Assorted (Su et al., 2025) mixes latent discrete tokens produced by a VQ-VAE with text tokens to shorten traces while maintaining accuracy. System1.5 (Wang et al., 2025) relies on a two-stage pipeline: first, a student model is aligned with a teacher model and subsequently a router is learned to encourage early exit via depth-wise and stepwise shortcuts. Our method is orthogonal, addressing the core challenge in latent reasoning, the absence of a direct supervision signal for these internal thoughts.

**KV-cache Compression.** KV-cache compression for reasoning focuses on trimming long, redundant thinking while preserving accuracy and throughput. R-KV (Cai et al., 2025) compresses on-the-fly by jointly scoring importance and redundancy to retain near-full performance with roughly 10–30% of the KV-cache on math reasoning, while KeyDiff (Park et al., 2025) offers a key-similarity–based eviction rule that preserves salient semantics under tight budgets. Other strategies such as HeadKV (Fu et al., 2025), PyramidKV (Cai et al., 2024), LESS (Dong et al., 2024), and Eigen Attention (Saxena et al., 2024), provide complementary reductions via head selection, hierarchical/pyramidal retention, importance-aware mixed-precision, and low-rank attention, yielding robust long-context and reasoning behavior. KV-Distill (Chari et al., 2025) instead learns a lightweight adaptor that compresses long-context KV-caches and trains a compressed-cache student to match a full-cache teacher via output-level KL alignment. In contrast, our proposed approach uses existing learning-free compression methods and treats the teacher's compressed KV-cache as supervision targets and distills them directly into the student's latent reasoning steps. As a result, the student model can directly generate the compressed KV-cache at inference time, bypassing an expensive step of generating the full CoT traces.

## 3 KAVA: KV-CACHE DISTILLATION FOR LATENT REASONING

### 3.1 OVERVIEW

We will split the common chat template into three parts named question Q, reasoning trace C and answer A, with $N_Q$, $N_C$ and $N_A$ token correspondingly. Consider an autoregressive generative model (LLM) that predicts each subsequent token conditioned on all preceding tokens. Latent reasoning introduces a set of unobserved intermediate steps, $Z = \{z_i\}_{i=1}^{M}$, which act as a substitute for the explicit reasoning trace C (see Fig. 2). The latent reasoning sequence begins with a special token `<bot>`, continues with $M$ *continuous tokens*, and terminates with `<eot>`, marking the end of the reasoning stage. During inference, these continuous *latent tokens* are generated by the same

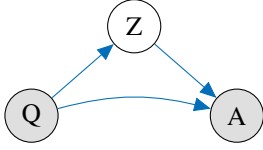

Figure 2: Graphical model of the latent reasoning generative model. The question prompt is used to generate continuous latent thought Z. The answer tokens are generated from the question and latent reasoning trace.

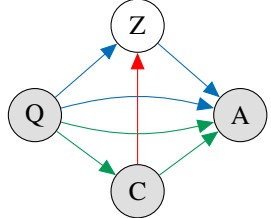

Figure 3: During training the student predicts the answer using latent tokens, teacher has the access to the full reasoning trace, and KV matching distills the information from the full to the latent CoT.

autoregressive model, bypassing the mapping of the embeddings to hard tokens. Instead, a (trainable) projection layer maps these continuous embeddings to the input embeddings that are used to predict the next token. We use the terms latent CoT and Continuous CoT (CCoT) interchangeably throughout the paper to refer to the tokens from Z.

**Training Objective.** Unlike chain-of-thought (CoT) reasoning traces, latent reasoning lacks direct supervision because latent traces are unobserved during training. Consequently, its performance is typically inferior to models trained with full CoT supervision (Deng et al., 2023; 2024). To address this, we leverage the observed reasoning traces C to guide latent reasoning during training, as illustrated in Fig. 3. This guidance is realized through distillation from teacher to student. Following Shen et al. (2025), we adopt a self-supervised framework in which the same model learns from explicit reasoning traces (as the teacher) as well as latent tokens (as the student).

We introduce KAVA, model with a novel objective, *KV-cache distillation*, to transfer relevant information from the teacher's reasoning trace to the student. An overview of this approach is depicted in Figure 1, with details provided in Section 3.2.

Our proposed KV-cache distillation loss is complementary to the CODI distillation loss introduced by Shen et al. (2025). CODI uses a single distillation token and matches its hidden activations between the teacher and the student models:

$$\mathcal{L}_{\text{CODI}} = \frac{1}{L} \sum_{l=1}^{L} \left| \text{sg}[\mathbf{h}_t^l] - \mathbf{h}_s^l \right|, \tag{1}$$

where $L$ is the total number of layers in the model, `sg` is a stop-gradient operator and $\mathbf{h}^l$ are model's hidden activation from layer $l$. The distillation token is chosen as the one preceding the answer. For example, if the answer is formatted as `"The answer is:5"`, the semicolon `":"` is used as the distillation token.

We combine KV-cache distillation with the CODI self-distillation to add a richer supervision signal to the latent reasoning trace. The total training objective is the following:

$$\mathcal{L}_{\text{KaVa}} = -\underbrace{\frac{1}{N_A} \log p(\mathbf{A}|\mathbf{Z}, \mathbf{Q})}_{\text{Student loss}} - \underbrace{\frac{1}{N_A + N_C} \log p(\mathbf{A}, \mathbf{C}|\mathbf{Q})}_{\text{Teacher loss}} + \underbrace{\alpha_1 \mathcal{L}_{\text{CODI}} + \alpha_2 \mathcal{L}_{\text{KV}}}_{\text{CODI and KV distillation}}, \tag{2}$$

where $\log p(\cdot)$ stands for cross-entropy loss, $\alpha_1$ and $\alpha_2$ are the hyperparameters that are used to balance the distillation terms, $N_A$ and $N_C$ denote number of tokens in the answer and CoT trace.

**Parallel Decoding.** Since latent tokens are generated sequentially, they do not allow for parallel decoding during training, which limits scalability. To mitigate this issue, we use Jacobi iteration over latent tokens to improve training and inference efficiency as proposed by Wu et al. (2025). Instead of generating latent tokens one by one during training PCCoT performs iterative updates of all tokens simultaneously for a predefined number of iterations $T$. PCCoT uses $T < M$, so that total number of forward passes is reduced from the number of latent tokens $M$ to the number of iterations $T$. For $T = M$ the method recovers the CODI explicitly and for $T = 0$ it corresponds to the Pause Token (Goyal et al., 2024).

## 3.2 KV-CACHE DISTILLATION

To provide an additional supervision signal from the full chain-of-thought (CoT) trace to the latent reasoning process, KAVA uses a distillation method based on matching the respective key-value (KV) caches (last term in Eq. 2). We apply redundancy-aware KV-cache compression to the teacher's cache prior to distillation. This encourages the student to generate compressed and abstract representations, while preserving crucial reasoning information from the CoT trace.

We first extract the KV-cache for both the explicit reasoning trace (teacher) and the latent thought (student). Each cache consists of key and value tensors for every token $i$, layer $l \in (1, \ldots, L)$, and attention head $h \in (1, \ldots, H)$ of the transformer:

$$K_t, V_t \in \mathbb{R}^{N_C \times H \times L \times d}, \qquad K_s, V_s \in \mathbb{R}^{M \times H \times L \times d}, \tag{3}$$

where $t$ stands for teacher and $s$ for the student. We use the last Jacobi iteration $T$ to extract the KV-cache of the student.

**Addressing the Length Mismatch.** The teacher cache $(K_t, V_t)$ and student cache $(K_s, V_s)$ differ in sequence length, since $M < N_C$. To align them while enforcing compression, we apply redundancy-aware KV eviction (Park et al., 2025; Cai et al., 2025) and obtain a compressed teacher cache $\widetilde{K}_t, \widetilde{V}_t \in \mathbb{R}^{M \times H \times L \times d}$. Specifically, we adapt R-KV (Cai et al., 2025) to select the top $M$ KV-pairs (see App. A) based on a combined redundancy–importance score $S_{i,h,l}$:

$$S_{i,h,l} = \lambda \underbrace{I_{i,h,l}}_{\text{Importance}} + (1 - \lambda) \underbrace{R_{i,h,l}}_{\text{Redundancy}}, \quad \lambda \in [0, 1], \tag{4}$$

where $\lambda$ is a hyperparameter controlling the balance between redundancy and importance. The eviction method is only applied during training, since the student is distilled to generate the compressed KV-cache. Since eviction method is not applied during inference, we leverage the answer tokens from the training data for the importance score computation. For each layer and head, we compute the attention score using the teacher's keys $K_t^{\cdot, h, l} \in \mathbb{R}^{N_C \times d}$ and queries corresponding to the answer tokens tokens $Q^{\cdot, h, l} \in \mathbb{R}^{N_A \times d}$:

$$A^{\cdot, \cdot, h, l} = \text{softmax}(Q^{\cdot, h, l} \cdot (K_t^{\cdot, h, l})^T) / \sqrt{d} \in \mathbb{R}^{N_A \times N_C}. \tag{5}$$

The importance score is then aggregated over all answer tokens[1]:

$$I_{\cdot, h, l} = \frac{1}{N_A} \sum_j A^{j, \cdot, h, l} \in \mathbb{R}^{N_C}. \tag{6}$$

Note that this computation incurs negligible overhead, since the attention scores were computed during the teacher's forward pass. Following R-KV[2], we compute a redundancy score $R_{i,h,l}$ as the average pairwise cosine similarity among all key vectors and normalize via softmax.

Finally, we use the score values $S_{i,h,l}$ (Eq. 4) to select top-$M$ keys (and their corresponding values) for each head and layer in the teacher's KV-cache. Full details and pseudocode are provided in App. A.

**KV Matching.** Independent KV-pair eviction across layers and heads alters the cache's structure and contents, yet it remains usable by the original model (see Figure 1b). However, there no longer exists a correspondence between the resulting cache and hard tokens. For that reason, we cannot apply standard ways of distillation, matching the activations of the teacher and student model. Instead, we propose distilling the keys and values directly.

To this end, we distill the latent reasoning cache to match the compressed teacher's cache, effectively guiding the latent model to approximate the full reasoning process in a more efficient and abstract form. We combine the loss for the keys and values in equal weights to get the final term of Eq. 2:

$$\mathcal{L}_{\text{KV}} = \frac{1}{2M} \left( \|\text{sg}[\widetilde{K}_t] - K_s\|_p^p + \|\text{sg}[\widetilde{V}_t] - V_s\|_p^p \right), \tag{7}$$

---

[1] For the group-query attention setting multiple queries are sharing the same key-value pair. In this case we apply `MaxPool` operation over the group before computing the importance score.

[2] Official R-KV implementation is available at `https://github.com/Zefan-Cai/R-KV`.

where $\|\cdot\|_p$ denotes an $L^p$-norm. That is, we have $L_1$ loss for $p = 1$ and MSE loss for $p = 2$. Note, that we first generate the whole student sequence with Jacobi iterations and then perform the distillation.

# 4 EXPERIMENTS

## 4.1 SETUP

We follow the experimental setup of Shen et al. (2025) and Wu et al. (2025) and extend the evaluation to more LLM families. Below we discuss the setup in more detail.

**Model.** We conduct experiments using the pretrained `LLaMA3.2-1b-Instruct`, `LLaMA3.2-3b-Instruct` and `Qwen2.5-0.5b-Instruct` (Grattafiori et al., 2024; Team, 2024) models and fine-tune them using LoRA (Hu et al., 2022). We follow Shen et al. (2025) and Wu et al. (2025) by using the same LoRA setup (rank 128 with alpha value 32 and dropout 0.1) for all the experiments. We employ PCCoT, the approach proposed by Wu et al. (2025), to generate latent thoughts; where 24 continuous latent tokens are generated in parallel with 3 iterations.

We fine-tune the models on `GSM8k-AUG`, `GSM8k-AUG-NL` (Deng et al., 2023), and `MetaMathQA` (Yu et al., 2024). The first two datasets are augmented versions GSM8k (Cobbe et al., 2021), containing `385k` training examples, with traces generated by GPT-4. `GSM8k-AUG` is then further processed by keeping only *equations* and removing all natural language from the traces. `MetaMathQA` contains `395k` training examples augmented from GSM8k and MATH (Hendrycks et al.), with traces generated by GPT-3.5-Turbo. We provide a detailed description of the datasets in Appendix B. For in-distribution evaluation, we assess all models on the test split of the original GSM8k dataset (Cobbe et al., 2021). For zero-shot evaluation, we assess model generalization on two benchmarks: GSM8k-Hard (Gao et al., 2023), SVAMP (Patel et al., 2021). We evaluate the model trained on MetaMathQA on in-distribution MATH500 (Hendrycks et al.) and out-of-distribution multiArith (Roy & Roth, 2015) and DeepMind-Mathematics (Saxton et al., 2019).

**Hyperparameters.** For our method, we conduct a hyperameter sweep over the learning rate, KV-cache distillation loss coefficient ($\alpha_2$), $L^p$ norm of the loss and the normalization method (layer-wise loss normalization or none). We choose the best-performing model on validation and run this setting with three random seeds. We report all hyperparameters in Appendix C.

We report the results of baseline approaches from Shen et al. (2025) and Wu et al. (2025) where possible. For the models not used in prior work, we take the hyperparameters from LLaMA3.2-1b, sweep over learning rates and report the result for the best performing model. We compare our method to CODI (Shen et al., 2025), PCCoT (Wu et al., 2025), Implicit CoT (iCoT) (Deng et al., 2024) and Coconut (Hao et al., 2024). We report the Full CoT performance as an upper bound and No-CoT as a lower bound.

**MetaMathQA** contains reasoning traces that are, on average, three times longer than those in the GSM8K-Aug-NL dataset, making it a significantly more challenging benchmark for latent reasoning. To illustrate the trade-off between efficiency and accuracy, we train models with varying the proportion of CoT tokens (from 20% to 100%) replaced by 24 latent thoughts. The remaining CoT tokens are incorporated into the student's loss function (Eq. 2) together with the answer tokens. We observed that the CODI distillation loss frequently introduces training instabilities in this configuration and therefore set $\alpha_1 = 0$ when training KaVa. For CODI and PCCoT baselines, we improve stability by removing the final sentence from the original CoT traces, following the recommended practice for GSM8K-Aug. All other hyperparameters are kept fixed and identical across KaVa and all baselines. Further details of the experimental setup are provided in Appendix G.

## 4.2 RESULTS

We report the average performance with standard error in Table 1. KAVA consistently outperforms the baselines. Importantly, we observe that KAVA has a lower drop in performance when switching from artificial `GSM8k-AUG` to a more realistic `GSM8k-AUG-NL` dataset. In the latter scenario, compression of the Full CoT trace would be more substantial as the traces are considerably longer,

Table 1: Test accuracy on in-distribution test dataset and zero-shot evaluation on out-of-distribution datasets. We use † to denote results copied from Shen et al. (2025) and Wu et al. (2025). We consider full CoT as an upper bound on the performance and denote **best** latent reasoning method in bold and second-best with the line. We denote out method as KAVA .

| Method | GSM8k-AUG | | | GSM8k-AUG-NL | | |
|---|---|---|---|---|---|---|
| | **GSM8k** | **GSM8k-Hard** | **SVAMP** | **GSM8k** | **GSM8k-Hard** | **SVAMP** |
| QWEN2.5 - 0.5B - INSTRUCT | | | | | | |
| FULL CoT | 50.6 | 12.6 | 54.3 | 48.5 | 12.6 | 57.3 |
| NO-CoT | 31.5 | 7.4 | 34.5 | 31.5 | 7.4 | 34.5 |
| CODI | 37.5 | 8.1 | 47 | 20.2 | 4.9 | 33.3 |
| PCCoT | 20.5 | 4.1 | 33 | 19.1 | 4.2 | 30.2 |
| KAVA (ours) | **46.9** ±1.4 | **10.8** ±0.1 | **50.6** ±0.4 | **44.4** ±1.8 | **10.2** ±0.4 | **46.5** ±0.1 |
| LLAMA3.2 - 1B - INSTRUCT | | | | | | |
| FULL CoT | 63.4 | 14.8 | 67.9 | 53.2 | 13.3 | 62.9 |
| NO-CoT | 33.2 | 7.4 | 41.4 | 33.1 | 7.7 | 41.4 |
| ICoT | 19.0† | 4.4† | 40.9† | 15.2† | - | - |
| COCONUT | 45.3† | 9.9† | 48.8† | 27.2† | - | - |
| CODI | 53.9 ±0.5 (55.6†) | 12.6 ±0.3 (12.8†) | 59.0 ±0.5 (61.1†) | 50.1 ±0.1 (49.7†) | 11.5 ±0.2 | 56.2 ±0.2 |
| PCCoT | 54.2 ±2.3 (53.35†) | **12.9** ±0.1 | 57.7 ±0.4 | 51.1 (50.72†) | 12.3 | 56.2 |
| KAVA (ours) | **56.5** ±0.4 | 12.7 ±0.1 | 58.9 ±0.5 | **55.7** ±0.4 | **12.8** ±0.2 | **58.6** ±0.3 |
| LLAMA3.2 - 3B - INSTRUCT | | | | | | |
| FULL CoT | 73.2 | 21.6 | 78.0 | 68.4 | 20.5 | 77.6 |
| NO-CoT | 41.7 | 10.5 | 56.9 | 41.7 | 10.5 | 56.9 |
| CODI | 61.0 | 15.0 | 72.4 | 55.9 | 13.6 | **70.1** |
| PCCoT | 54.7 | 13.5 | 69.5 | 47.6 | 11.0 | 65.2 |
| KAVA (ours) | **65.7** | **15.2** | **72.7** | **60.0** | **14.8** | 66.1 |

Table 2: We measure the efficiency of different reasoning model by the average number of forward passes required to generate the reasoning trace and answer. We use † to denote results copied from Shen et al. (2025) and Wu et al. (2025). We report the improvement in efficiency compared to the Full CoT in (parentheses).

| Method | GSM8k-AUG | | | GSM8k-AUG-NL | | |
|---|---|---|---|---|---|---|
| | **GSM8k** | **GSM8k-Hard** | **SVAMP** | **GSM8k** | **GSM8k-Hard** | **SVAMP** |
| QWEN2.5 - 0.5B - INSTRUCT | | | | | | |
| FULL CoT | 40.4 | 59.6 | 23.3 | 82.4 | 105.2 | 44.9 |
| NO-CoT/ ICoT | **7.4** | **10.1** | **7.0** | **7.4** | **10.1** | **7.0** |
| CODI / COCONUT | 14.4 | 20.7 | 14.1 | 14.0 | 19.0 | 13.4 |
| KAVA (ours) / PCCoT | 9.5 (-76%) | 13.3 (-78%) | 8.9 (-62%) | 9.2 (-89%) | 13.5 (-87%) | 9.0 (-80%) |
| LLAMA3.2 - 1B - INSTRUCT | | | | | | |
| FULL CoT | 31.9 | 41.3 | 17.8 | 71.9 | 80.2 | 40.6 |
| NO-CoT / ICoT | **6.2** | **7.3** | **6.2** | **6.2** | **7.3** | **6.2** |
| CODI / COCONUT | 11.9 | 13.9 | 11.5 | 11.8 | 13.9 | 11.3 |
| KAVA (ours) / PCCoT | 6.9 (-78%) | 9.1 (-78%) | 6.5 (-63%) | 7 (-90%) | 10 (-88%) | 6.4 (-86%) |
| LLAMA3.2 - 3B - INSTRUCT | | | | | | |
| FULL CoT | 31.6 | 40.3 | 17.0 | 75.2 | 32.9 | 38.3 |
| NO-CoT / ICoT | **6.1** | **7.4** | 6.1 | 6.1 | **7.4** | 6.1 |
| CODI / COCONUT | 11.5 | 14.2 | 11.0 | 11.1 | 13.1 | 10.7 |
| KAVA (ours) / PCCoT | 6.4 (-80%) | 8.2 (-80%) | **6** (-65%) | **6** (-92%) | 7.9 (-76%) | **5.7** (-85%) |

while questions are kept the same. This demonstrates the better scalability of our approach. We additionally present these result in the form of Accuracy-Efficiency Pareto frontier in the Appendix F.

We also measure the efficiency of the method by the number of forward passes a model makes to generate the reasoning trace and the answer, reported in Table 2. We group the methods based on their inference behavior. For example, No-CoT and iCoT do not produce reasoning tokens, while CODI and Coconut rely on the same number of latent steps. KAVA builds on top of PCCoT, where we only use $T = 3$ iterations (forward passes) to generate all the latent tokens. For that reason, we group PCCoT with KaVA. Our method achieves better efficiency than CoT, requiring between 62% and 92% fewer forward passes per question compared to Full CoT.

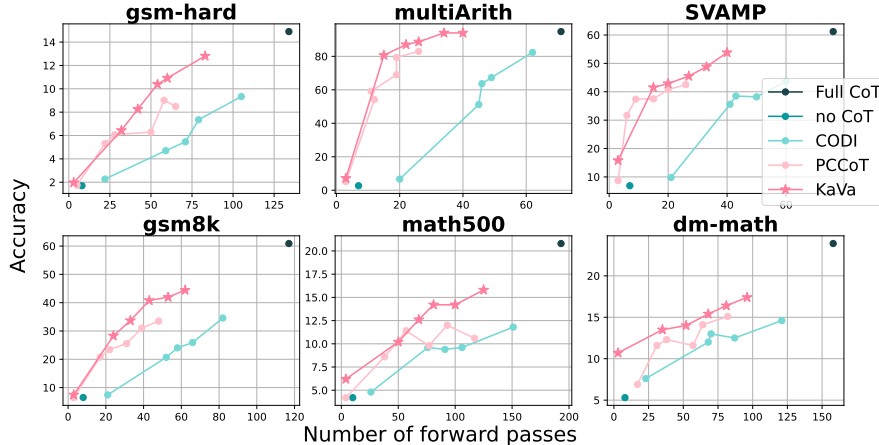

Figure 4: Llama-1b performance on MetaMathQA with varying latent reasoning ratios. During training, between 20% and 100% of reasoning tokens are replaced with 24 latent tokens; each point represents a model trained with a different replacement ratio. Retained tokens are split 10% before and 90% after the latent step.

Table 3: Test accuracy on GSM8k dataset without projection layer and distillation loss ($\alpha_1 = 0$).

| $\mathcal{L}_{KD}$ | PRJ. | GSM8k | GSM-Hard | SVAMP |
|---|---|---|---|---|
| ✓ | ✓ | **56.5** ±**0.4** | **12.7** ±**0.1** | **58.9** ±**0.5** |
| ✗ | ✓ | 52.8 ±0.1 | 12.2 ±0.1 | 56.2 ±0.2 |
| ✓ | ✗ | 52.2 ±0.6 | 12.3 ±0.2 | 58.3 ±0.3 |

Table 4: Test accuracy on GSM8k dataset when the teacher is trained on all the steps.

| $\mathcal{L}_{KD}$ | $\mathcal{L}_{KV}$ | Drop Last | All Steps |
|---|---|---|---|
| ✓ | ✓ | **56.5** ±**0.4** | 51.2 ±0.8 |
| ✓ | ✗ | *53.35* ±*0.18* | 47.2 ±2.9 |

**MetaMathQA results** Figure 4 illustrates the accuracy–efficiency trade-off for models trained on MetaMathQA across different replacement ratios. The leftmost dot at each curve corresponds to the model where 100% of the CoT tokens are replaced with the latent tokens, while subsequent points represent 60%, 50%, 40%, 30% and 20% replacement. In all cases, we replace tokens from the middle of the CoT trace, placing 10% of the retained tokens before the latent reasoning step. We provide additional results where beginning of the reasoning trace is replaced with the latent tokens in Appendix G. These curves highlight that KaVa consistently outperforms CODI and PCCoT across the entire Pareto frontier, offering superior accuracy at different efficiency levels.

## 4.3 ABLATION STUDIES

We select LLAMA3.2-1B-INSTRUCT to conduct ablation studies for our method. We run each experiment with three random seeds and report average test accuracy.

**Model Components.** First, we study how different modeling choices influence the final performance. In Table 3 we report benchmark performance when trained without the distillation loss (Shen et al., 2025) or without projection layer. As can be seen, both components are quite crucial, but even without them the method considerably outperforms the no-CoT baseline.

**Removing Last Step of the Trace.** Following Shen et al. (2025); Wu et al. (2025) we remove the last step from the teacher's reasoning trace. CODI demonstrates that this step is crucial for model performance, since otherwise the token that CODI chooses for distillation tends to be less informative. In Table 4 we train our model (using both KV matching and distillation) and PCCoT (only distillation) on all steps. Performance of our method drops much lower, indicating that KV-cache distillation loss compensates for the lack of usefulness of a distillation token in a fully automatic manner.

**KV Loss Sensitivity.** Matching keys and values of the KV-cache is a non-standard way of distillation. Therefore, we study the model sensitivity to the distillation loss type and coefficient. In Figure 5 we plot the test accuracy for two losses and three different coefficients. The model per-

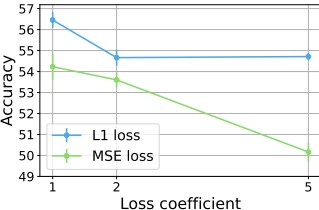 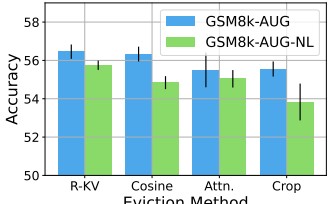 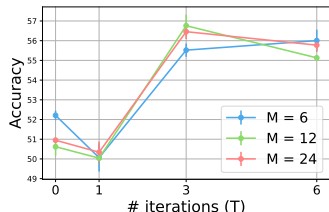

Figure 5: Test accuracy (%) of KAVA for different KV matching coefficient and loss.

Figure 6: Test accuracy (%) of KAVA with different eviction methods.

Figure 7: Test accuracy (%) of KAVA with different number of iterations and latent tokens.

forms consistently better with $L_1$ loss when trained on GSM8k-AUG and with Llama-1b. However, we observed that better performance may be achieved when using MSE loss on other datasets (see Appendix C for the detailed hyperparameters used for all models and datasets).

**KV Eviction.** We follow Cai et al. (2025) in using $\lambda = 0.1$ (see Eq. 4) in R-KV eviction for all the experiments. As an ablation study we consider the two extremes: cosine-only ($\lambda = 0$) and attention-only ($\lambda = 1$). These cases correspond to choosing the keys and values based on diversity or importance only. Furthermore, we use a simple baseline of cropping the full CoT trace from the right, that is we only keep first $M$ tokens of the teacher's cache for distillation. We report the results in Figure 6. We observe that combining both attention-based and similarity-based criteria enhances the performance for both datasets.

**Number of Tokens and Iterations.** Similarly to Wu et al. (2025), we observe that the number of iterations can have a different impact on accuracy depending on the number of latent tokens (Fig. 7). For larger numbers of latents (12, 24) we observe reduced performance beyond a certain number of iterations.

**Amount of Training Data.** We measure the impact of data scaling by training the Llama-1b on the GSM8k-Aug subsampled to 50% and 25% of the original size, finding that the amount of the training data is crucial for our method's performance (see App. H).

## 5 INTERPRETABILITY OF LATENT REASONING TRACES

### 5.1 DECODING THE LATENT TRACE

Although the latent CoT is not directly interpretable, one can still attempt to decode the reasoning trace from latent tokens. A straightforward approach is to project the final hidden state of the latent tokens via the language modeling head. An example of a decoded trace is shown in Table 5. More examples of the decoded traces are given in the Appendix E. Interestingly, the decoded latent trace is often identical to the trace generated by the teacher model, underlining the importance of the teacher guidance. In particular cases, as shown in the table, a reasoning step can be expressed in two equivalent forms (e.g. `<<650*2=1300>>` and `<<2*650=1300>>`). In regular CoT, this ambiguity is resolved after sampling a unique prefix of one of the variants, however, there is no explicit mechanism allowing for such resolution in a latent CoT. Nevertheless, the student arrives at the correct answer.

Models trained on the GSM8k-AUG dataset tend to produce latent CoT's that are easily interpretable. In contrast, models trained on the GSM8k-AUG-NL dataset resist this straightforward read-out method. We hypothesize that this is caused by the KV-cache distillation employed by KAVA —in a dataset with shorter traces, such as GSM8k-AUG, most of the time the KV-cache retains all of its content after eviction. On longer traces, such as the ones found in GSM8k-AUG-NL, not all content of the KV-cache is preserved, and, furthermore, each latent thought's distillation target may consist of keys and values originating from different tokens of the teacher's CoT. This can prevent latent thought to hard token correspondence from arising.

### 5.2 TEACHER-STUDENT KV-CACHE CORRESPONDENCE

We compute the cosine similarity of the keys and values in the latent CoT with (1) the ground truth KV-cache, and (2) the ground truth KV-cache after eviction. The results, averaged over attention

Table 5: Decoding the latent thoughts. A validation prompt is used: "Mrs. Taylor bought two smart televisions that cost \$650 each. If the total sales price had a 25% discount, how much did Mrs. Taylor pay for the two televisions?". Latent thoughts 16-24 are not shown due to their limited semantic value. 3 tokens with the highest logits are shown for each latent thought. Tokens `T1`, `T2`, `T3`, `T4`, `T5`, `T6`, `T7` stand for `␣total`, `␣cost`, `␣dollars`, `␣discount`, `␣original`, `␣gross`, and `␣price` respectively. Following CODI, the teacher is trained on traces omitting the last step.

| TopK | 1 | 2 | 3 | 4 | 5 | 6 | 7 | 8 | 9 | 10 | 11 | 12 | 13 | 14 | 15 | Answer |
|---|---|---|---|---|---|---|---|---|---|---|---|---|---|---|---|---|
| | | | | | | GSM8K-Aug | | | | | | | | | | |
| 1 | 650 | * | 2 | = | 130 | 0 | >> | << | ␣of | 0 | * | * | >> | = | = | |
| 2 | 2 | + | 650 | * | 650 | >> | . | The | . | * | % | % | = | * | 325 | 975 |
| 3 | 65 | – | 0 | =\$ | 125 | 00 | \| | <<( | ␣and | k | *. | = | 0 | ␣= | 125 | |
| Teacher | | | | | | <<650*2=1300>><<1300*25/100=325>> | | | | | | | | | | 975 |
| Golden | | | | | <<650*2=1300>> <<1300*25/100=325>><<1300–325=975>> | | | | | | | | | | | 975 |
| | | | | | | GSM8K-Aug-NL | | | | | | | | | | |
| 1 | T1 | ␣of | ␣of | 0 | ␣\$ | ␣\$ | ␣\$ | ␣\$ | ␣\$ | ␣\$ | ␣\$ | ␣ | ␣ | T4 | T4 | |
| 2 | T2 | T2 | T2 | T3 | \$ | \$ | \$ | \$ | \$ | \$ | \$ | ␣\$ | T4 | ␣ | ␣ | 975 |
| 3 | T5 | T7 | ␣was | T6 | ␣was | ␣ | ␣ | ␣ | ␣ | ␣ | ␣ | \$ | The | , | , | |
| Teacher | The total cost of the two televisions is 2 x \$650 = \$1300 [...] \$1300 x 25/100 = \$325. | | | | | | | | | | | | | | | 975 |
| Golden | The total cost of the two smart televisions is [...] \$975 for the two smart televisions. | | | | | | | | | | | | | | | 975 |

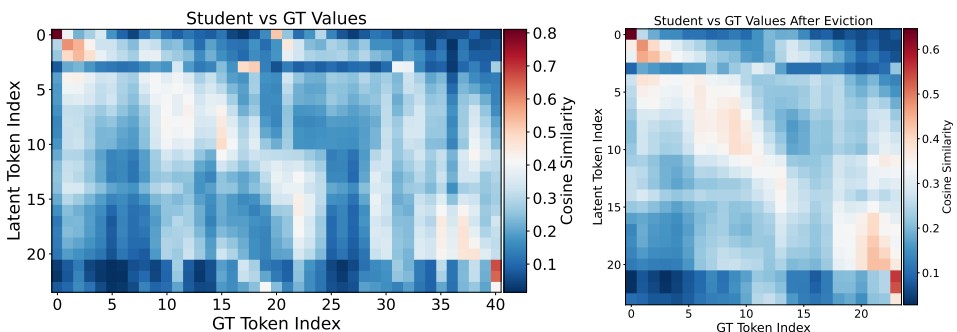

Figure 8: Cosine similarity between attention Values in the latent CoT and the ground truth CoT, averaged across all heads and layers. We use the same prompt and ground truth CoT as in Table 5.

heads and layers, are presented in Figures 8, 9. We observe that when comparing to the KV-cache after eviction, the similarities near the diagonal tend to be higher, which is expected, as it is encouraged by the KV distillation. Furthermore, the values to the right of the diagonal are higher when comparing with the full CoT, which is desired, as this represents the compression of the original CoT (i.e. the key of a $n$-th latent token is similar to the key of an $m$-th hard token where $n < m$). The full visualization of the similarities across layers and heads can be found in the App. D.

## 6 CONCLUSION AND DISCUSSION

We introduce KAVA, a novel framework that bridges the supervision gap in latent reasoning by distilling knowledge from a teacher model's compressed Key-Value (KV) cache. Our central contribution is the demonstration that a compressed KV-cache, despite losing direct token correspondence, can serve as a rich, stepwise supervisory signal for a latent reasoning student. By aligning the student's latent trajectory with the teacher's internal reasoning dynamics in KV space, KAVA overcomes the limitations of token-level distillation and the inefficiencies of verbose Chain-of-Thought (CoT) traces. KAVA consistently outperforms strong latent reasoning baselines, scales effectively to larger backbones, and shows robust performance on natural-language reasoning datasets where prior methods often struggle. While the advancement of latent reasoning is linked to the availability of large-scale training data to instill novel reasoning dynamics, our work establishes compressed KV-cache distillation as a scalable and effective supervision technique for developing efficient and powerful reasoning models.

## 7 ETHICS AND REPRODUCIBILITY STATEMENTS

We adhere to the ICLR Code of Ethics. During the preparation of this manuscript, we utilized large language models (LLMs) to assist with grammar correction and refinement of the writing.

Regarding the interpretability of reasoning traces, a gap exists between latent reasoning and CoT-based reasoning. It should however be noted that in light of recent findings (Schoen et al., 2025), even CoT should not be considered a fully interpretable method. Furthermore, an argument can be made that due to the distillation used in our method and other latent approaches (Shen et al., 2025; Wu et al., 2025), the safety risk posed by current latent reasoners is, in fact, lesser than in the case of more powerful CoT reasoners. We believe it is important that future work focus on explaining the reasoning in both latent and non-latent approaches as well as on mitigating any threats posed by increasingly capable models.

In this paper, we provide all the necessary details to ensure the reproducibility of the presented method. We describe our method in Section 3 and provide pseudocode and method details in Appendix A. We provide training protocols in Section 4, all the hyperparameters in the Appendix C, and data description in Appendix B.

### ACKNOWLEDGMENTS

We gratefully acknowledge Polish high-performance computing infrastructure PLGrid (HPC Centers: ACK Cyfronet AGH, WCSS) for providing computer facilities and support within computational grant no. PLG/2025/018134.

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

# A  KV EVICTION DETAILS

We provide pseudocode to compute the r-KV score in Listing 1. The function takes as input a key-value pair and the attention scores between the CoT and and Answer tokens. There are several implementation differences from the original R-KV method.

**Padding Tokens**  First, we need to take into account padding tokens since we evict KV-cache in a batch during training. We do that by always assigning the lowest possible redundancy and importance score to the value-key pairs corresponding to the padding tokens

**Importance Score**  To compute the importance score, we use the attention score that answer tokens get when attending to the full CoT. We extract those value during the normal teacher forward pass and reuse to compute the

**Retention of Recent Tokens**  R-KV implementation adjust the redundancy score by always keeping $\beta$ the most recent tokens. This is important for a reliable model performance during generation. We only use our method during training and apply it to the whole reasoning trace, therefore we skip this adjustment and only rely on selecting the most diverse keys with high attention to the answer tokens.

Listing 1: Pseudocode to implement the eviction score for a given key-value pair.

```
def r_kv_score(key: torch.tensor, attn: torch.tensor, lbd: float):
    """
    key: torch.tensor [bs, N_c, d] - CoT keys for a single head and layer
    attn: torch.tensor [bs, N_A, N_c] - attenton scores
    lbd: float - the weight of the importance score
    """
    # compute redundancy score
    key_norm = key / (key.norm(dim=-1, keepdim=True) + 1e-8)
    cosine_sim = torch.einsum("...id,...jd->...ij", key_norm, key_norm)
    for i in range(cosine_sim.shape[0]):
        cosine_sim[i].fill_diagonal_(0)
    cos_score = torch.sum(-cosine_sim, dim=-2) / torch.sum(
        ~pad_tokens, dim=-1, keepdim=True
    )
    # Normalize to 1
    R = cos_score.softmax(dim=-1)
    pad_tokens = key.sum(-1) == 0
    R[pad_tokens] = 0

    # compute importance score
    # sofmax over CoT dimention and avrage over answer tokens
    I = F.softmax(attn, dim=-1).mean(-2)
    # Assign the lowest score to the padding tokens
    I[pad_tokens] = 0

    S = lbd * I + (1 - lbd) * R
    return S
```

# B  DATASETS

Our models are trained using the GSM8k-Aug and GSM8k-Aug-NL datasets introduced by
Deng et al. (2023), which augment the training set of the GSM8k (Cobbe et al., 2021)
using GPT4 and provide a separate validation split.  The golden traces in the datasets
are split into discrete steps.  GSM8k-Aug traces consist only of succinct statements such
as `<<600*30/100=180>>`; `<<600*10/100=60>>`.  The questions and answers in the
NL (Natural Language) subset are identical, however the steps are formulated in natural
language: `600 x 30/100 = 180 employees were promoted.`; `600 x 10/100 =
60 employees received a bonus.`

|  | GSM8K-Aug | GSM8K-Aug-NL | MetaMathQA |
|---|---|---|---|
| Huggingface Path | whynlp/gsm8k-aug | whynlp/gsm8k-aug-nl | meta-math/MetaMathQA |
| No. of Train Sample | | 385,620 | 395,000 |
| No. of Valid. Samples | | 500 | - |
| No. of Test Samples | | 1319 | 1319 + 500 |
| Average CoT len | 23.1 | 55.0 | 148.9 |

## C  HYPERPARAMETERS

Table 6: All the hyperparameters used for our method.

| Hyperparameter | GSM8k-AUG | GSM8k-AUG-NL | MetaMathQA |
|---|---|---|---|
| | LLAMA3.2 - 1B - INSTRUCT | | |
| $\alpha_1$ (CODI) | 10 | 10 | 0 |
| KV loss | Smooth L1 | MSE | L1 |
| Layer-wise std | True | True | False |
| $\alpha_2$ (KV) | 1 | 1 | 1 |
| r-kv $\lambda$ | 0.1 | 0.1 | 0.1 |
| Use Projection | True | True | True |
| learning rate | 8e-4 | 8e-4 | 8e-4 |
| lr scheduler | Cosine | Cosine | Cosine |
| optimizer | AdamW | AdamW | AdamW |
| batch size | 128 | 128 | 64 |
| weight decay | 0.1 | 0.1 | 0.1 |
| gradient clipping | 2 | 2 | 2 |
| epochs | 10 | 10 | 5 |
| | QWEN2.5 - 0.5B - INSTRUCT | | |
| $\alpha_1$ (CODI) | 10 | 10 | |
| KV loss | MSE | MSE | |
| Layer-wise std | False | True | |
| $\alpha_2$ (KV) | 1 | 1 | |
| r-kv $\lambda$ | 0.1 | 0.1 | |
| Use Projection | True | True | |
| learning rate | 5e-4 | 8e-4 | |
| lr scheduler | Cosine | Cosine | |
| optimizer | AdamW | AdamW | |
| batch size | 128 | 128 | |
| weight decay | 0.01 | 0.1 | |
| gradient clipping | 2 | 2 | |
| epochs | 10 | 10 | |
| | LLAMA3.2 - 3B - INSTRUCT | | |
| $\alpha_1$ (CODI) | 20 | 20 | |
| KV loss | Smooth L1 | Smooth L1 | |
| Layer-wise std | False | False | |
| $\alpha_2$ (KV) | 2 | 2 | |
| r-kv $\lambda$ | 0.1 | 0.0 | |
| Use Projection | True | False | |
| learning rate | 2e-4 | 2e-4 | |
| lr scheduler | Cosine | Cosine | |
| optimizer | AdamW | AdamW | |
| batch size | 128 | 128 | |
| weight decay | 0.1 | 0.1 | |
| gradient clipping | 2 | 2 | |
| epochs | 5 | 5 | |

# D   KV-CACHE COSINE SIMILARITY BETWEEN THE LATENT COT AND THE GROUND-TRUTH COT

We investigate the similarity between the KV-cache representing the latent CoT and the KV-cache of the ground-truth CoT. Figures 8 and 9 present the similarities averaged over layers and heads, while figures 10, 11, 12, and 13 show the similarities in individual heads and layers.

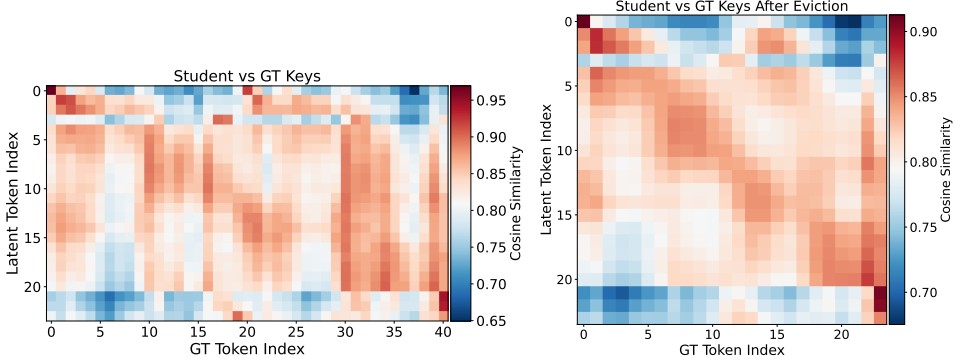

Figure 9: Cosine similarity of Keys in the latent CoT with Keys of the ground truth averaged across heads and layers. We use the same prompt and ground truth CoT as in Table 5.

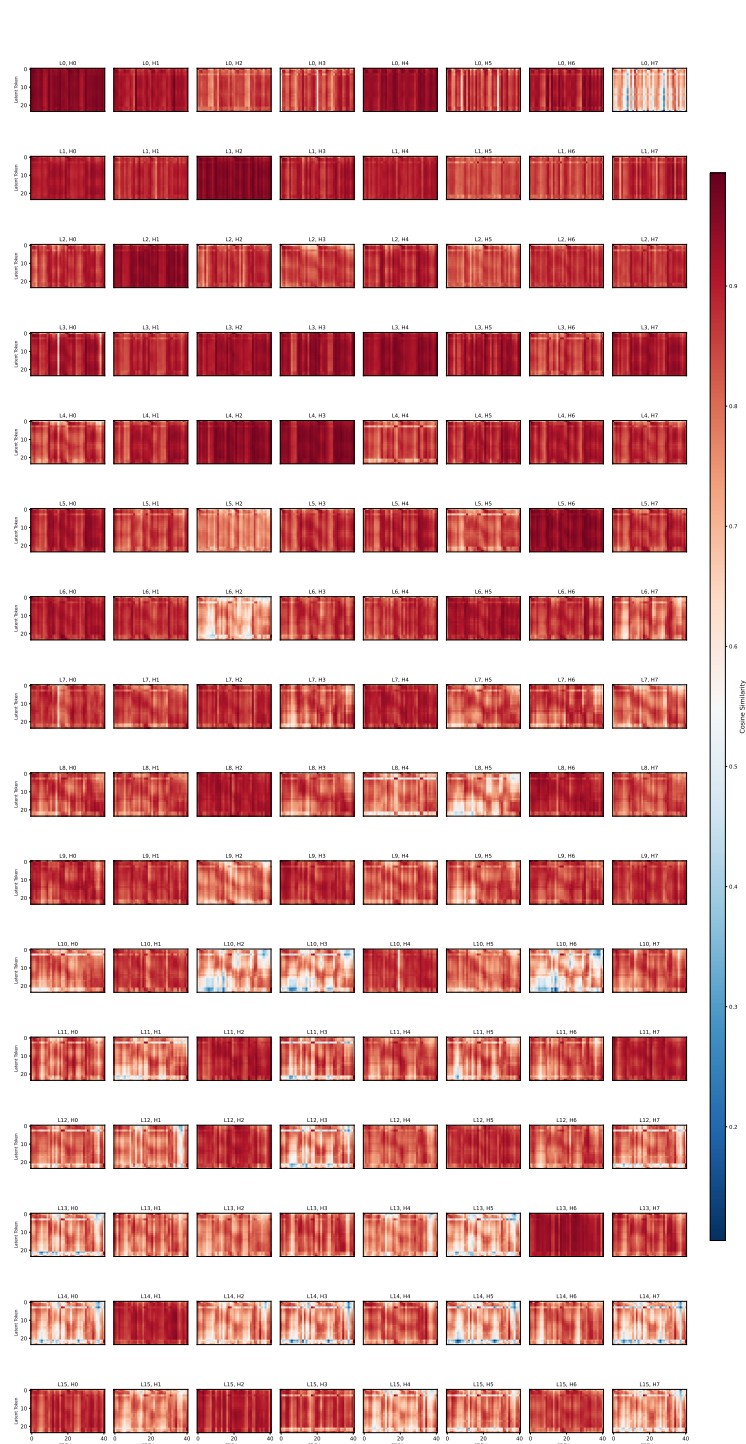

Figure 10: Cosine similarity between Keys in the latent CoT and Keys of the ground truth across layers.

Figure 11: Cosine similarity between Values in the latent CoT and Values of the ground truth across layers.

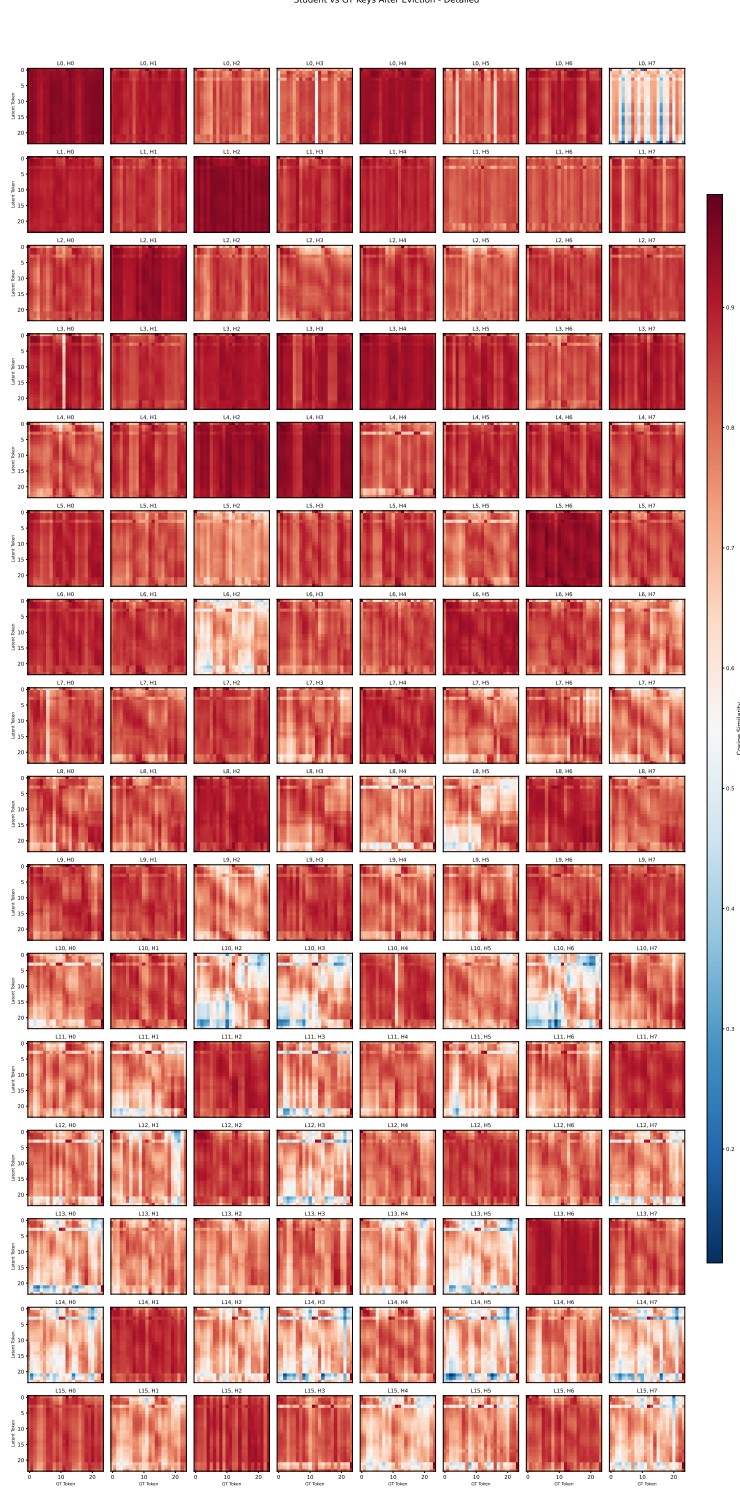

Figure 12: Cosine similarity between Keys in the latent CoT and Keys of the ground truth after eviction across layers.

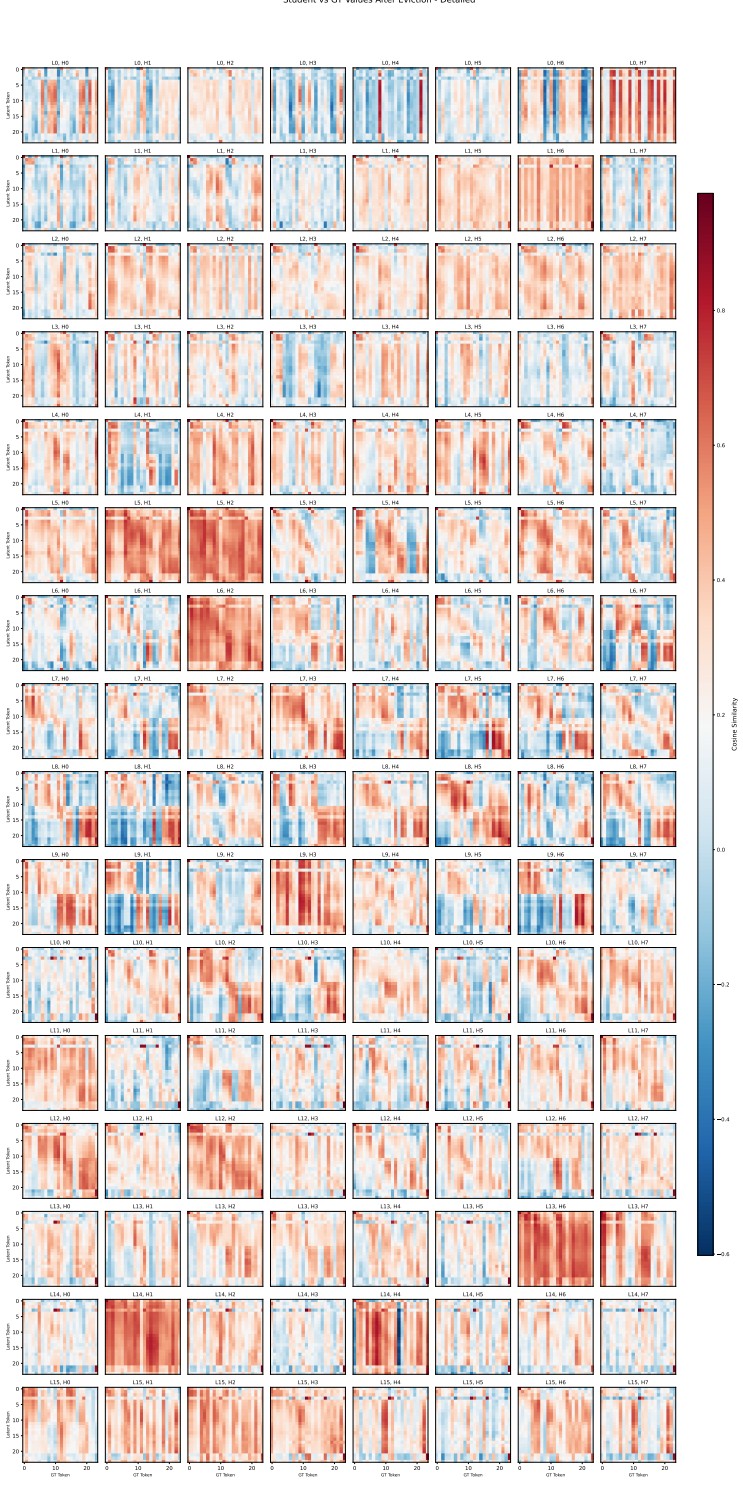

Figure 13: Cosine similarity between Values in the latent CoT and Values of the ground truth after eviction across layers.

## E  DECODED LATENT TRACES

In this section we present two additional examples of traces decoded in the same manner as described in section 5.1.

| TopK | 1 | 2 | 3 | 4 | 5 | 6 | 7 | 8 | 9 | 10 | 11 | 12 | 13 | 14 | 15 | Answer |
|---|---|---|---|---|---|---|---|---|---|---|---|---|---|---|---|---|
| | | | | | | GSM8K-Aug | | | | | | | | | | |
| 1 | 24 | * | 50 | = | 120 | 0 | >> | << | 120 | * | 0 | 0 | 0 | = | = | |
| 2 | 50 | *. | 0 | * | 150 | >> | . | The | 0 | *. | *. | 10 | >> | >> | 0 | 3600 |
| 3 | . | *( | 30 | *. | 600 | 00 | << | <<( | . | 0 | * | 00 | 00 | 0 | >> | |
| Teacher | | | | | | <<50*0.10=5>><<5*24=120>> | | | | | | | | | | 3600 |
| Golden | | | | | | <<50*.10=5>><<5*24=120>><<120*30=3600>> | | | | | | | | | | 3600 |
| | | | | | | GSM8K-Aug-NL | | | | | | | | | | |
| 1 | T6 | ␣ | 50 | T9 | * | * | ␣ | ␣ | ␣ | ␣ | , | , | , | , | 0 | |
| 2 | T7 | T6 | 0 | 0 | ␣ | ␣ | * | * | * | , | T11 | T10 | T10 | T10 | ␣per | 3600 |
| 3 | T8 | ␣a | * | * | T11 | T11 | T11 | T11 | , | * | ␣ | ␣per | ␣per | ␣ | 00 | |
| Teacher | | | | | | He gets 0.10*50=5 dollars a hour | | | | | | | | | | 1800 |
| Golden | | | | | | He makes 50*$.10=$5 per hour [...] $120*30=$3600 a month | | | | | | | | | | 3600 |

Table 7: Prompt: "Jon runs a website where he gets paid for every person who visits. He gets paid $0.10 for every person who visits. Each hour he gets 50 visits. His website operates 24 hours a day. How many dollars does he make in a 30 day month?". T6 – T11 stand for ␣gets, ␣makes, ␣operates, ␣visits, ␣hourly, and ␣hour respectively. Tokens 16-24 are omitted due to low semantic content.

| TopK | 1 | 2 | 3 | 4 | 5 | 6 | 7 | 8 | 9 | 10 | 11 | 12 | 13 | 14 | Answer |
|---|---|---|---|---|---|---|---|---|---|---|---|---|---|---|---|
| | | | | | | GSM8K-Aug | | | | | | | | | |
| 1 | 150 | * | 2 | = | 300 | >> | The | ␣as | ␣as | ␣as | ␣as | ␣as | ␣as | ␣as | |
| 2 | 2 | + | 1 | * | 150 | . | << | T15 | T15 | T15 | T15 | T15 | T15 | T15 | 1500 |
| 3 | 300 | ␣* | 5 | ␣= | 30 | ␣>> | T16 | ␣of | ␣of | ␣of | ␣of | ␣of | ␣of | ␣of | |
| Teacher | | | | | | <<150*2=300>> | | | | | | | | | 1500 |
| Golden | | | | | | <<150*2=300>><<300*5=1500>> | | | | | | | | | 1500 |
| | | | | | | GSM8K-Aug-NL | | | | | | | | | |
| 1 | T13 | T11 | T11 | T17 | T11 | T11 | T11 | T11 | T11 | T11 | T11 | T11 | T11 | T11 | |
| 2 | T11 | ␣to | T14 | T12 | ␣to | T14 | T14 | ␣ | ␣ | ␣ | ␣ | ␣ | T14 | T14 | 1500 |
| 3 | T14 | T18 | ␣to | T11 | T14 | ␣to | ␣ | T14 | T14 | T14 | T14 | T14 | , | , | |
| Teacher | | | | | Raine takes 150 x 2 = 300 steps walking to and from school in one day. | | | | | | | | | | 1500 |
| Golden | | | | Raine takes 150 x 2 = 300 steps walking [...]  her 300 x 5 = 1500 steps in five days. | | | | | | | | | | | 1500 |

Table 8: Prompt: "Raine's house is just a walking distance from her school. It takes her 150 steps to walk to the school. How many steps does she take walking to and from school in five days?". T11 – T18 stand for ␣walking, ␣footsteps, ␣walks, ␣walk, ␣but, This, ␣steps, and ␣going respectively.

# F    ACCURACY-EFFICIENCY TRADE-OFF

In Figures 14, 15 we plot results from the Tables 1, 2 in the form of the Pareto frontier, where the closer model us to the **top left** corner, the better. In most cases KAVA exhibit the best accuracy-efficiency trade-off. Note that some baseline approaches do not report number of generated tokens or forward passes, making it impossible to add them to this plots. In this case, we only compare in term of accuracy in Table 1.

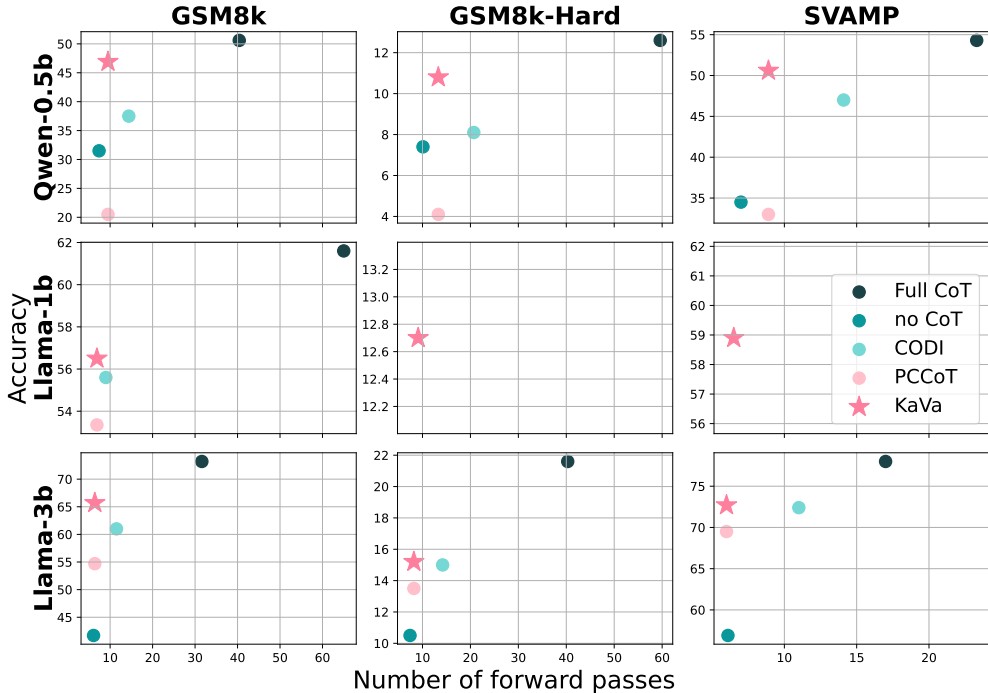

Figure 14: Accuracy-Efficiency trade-off of KAVA and baseline approaches trained on `GSM8k-Aug` dataset for three different model architectures. KAVA consistently demonstrates better trade-off, by being closer to the **top-left** corner.

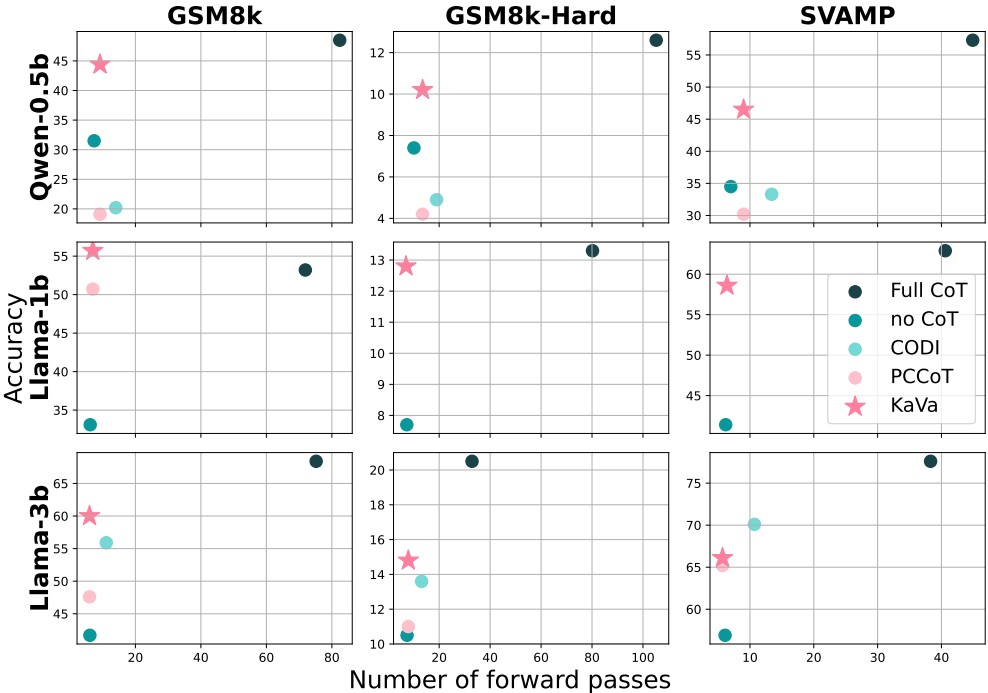

Figure 15: Accuracy-Efficiency trade-off of KAVA and baseline approaches trained on `GSM8k-Aug-NL` dataset for three different model architectures. KAVA consistently demonstrates better trade-off, by being closer to the **top-left** corner.

## G METAMATHQA EXPERIMENTS

In our initial experiments, we observed that latent reasoning methods only slightly outperform the no CoT baseline when the entire CoT trace is replaced by latent reasoning (see the bottom-left points in Figures 16 and 4). We hypothesize two main reasons for this: (1) distilling a long reasoning trace into a short latent trajectory is challenging because the number of latent tokens remains fixed, and (2) each question in the MetaMathQA dataset (unlike in GSM8k-Aug) is associated with multiple reasoning traces, reducing the effective number of unique questions compared to the dataset size.

**Hybrid Reasoning**  To address this limitation, we introduce hybrid reasoning, where a portion of the original reasoning trace is retained as hard tokens. This approach alleviates both constraints: (1) fewer tokens are replaced, and (2) the student model retains part of the original trace in its objective, increasing training data diversity.
We report results for varying proportions of the original trace being replaced $(20\%, 30\%, 40\%, 50\%, 60\%)$ in a form of a Pareto curve. The leftmost point on each curve represents the non-hybrid setting, where all hard tokens are replaced. We only report results for non-diverged runs for CODI and PCCoT.

**Location of the latent tokens**  We evaluated two strategies for positioning latent tokens: at the beginning of the reasoning process and in the middle. Results for the beginning placement are shown in Figure 16, and for the middle placement in Figure 4. In both settings, KAVA consistently outperforms all baselines.

We trained KAVA with KV-distillation only, setting CODI distillation loss coefficient to 0 ($\alpha_1 = 0$). This way KAVA trianing remained stable when scaling to longer reasoning traces or using a hybrid approach, unlike CODI and PCCoT, which often diverged, especially when replacing the beginning of the reasoning trace. Removing the final CoT sentence (per CODI's recommendation) improved but did not fully resolve instability, so we report only non-diverged runs.

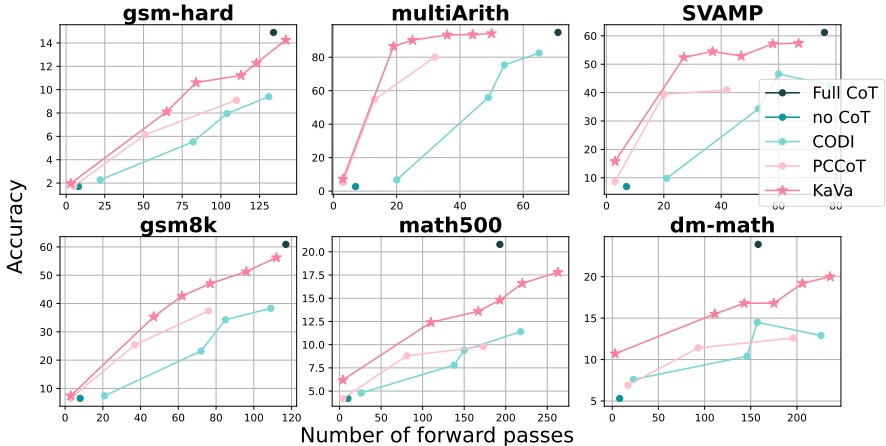

Figure 16: Test results for Llama-1b model trained on MetaMathQA dataset. During training, between 20% and 100% of reasoning tokens are replaced with 24 latent tokens; each point represents a model trained with a different replacement ratio. When non-replaced tokens remain, they are placed after the latent reasoning step.

## H TRAINING DATASET SIZE

We performed an assessment of the impact of the size of the dataset on the performance of our method. We find that the size of the training dataset plays a crucial role for KaVa – when trained on a fraction of the dataset, the performance suffers and cannot be recovered even if the number of total training steps is matched with training on the full dataset. In fact, we see 10 epochs is the optimal amount of training epochs (among the candidate values), regardless of the dataset size.

As shown in the bottom plots of Figure 17, PCCoT exhibits a more pronounced decline in accuracy than KaVa when the amount of training data is reduced. While this degradation can be partially mitigated by increasing the number of training epochs, KaVa consistently outperforms PCCoT even when the total number of training iterations is matched to the original setting.

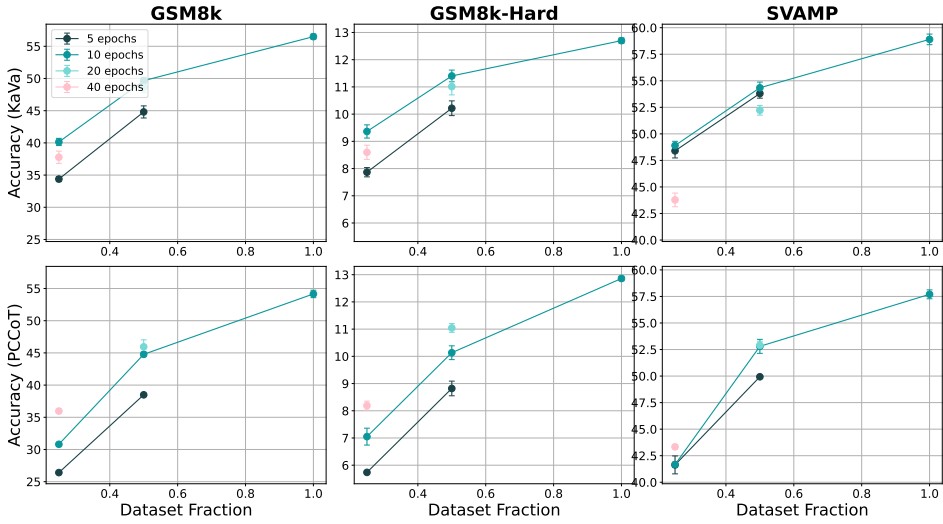

Figure 17: The impact of the size of the dataset on the performance of KaVa (top) and PCCoT (bottom). 10 epochs matches the number of epochs in all the experiments, whereas 20 and 40 epochs match the total number of training steps when training on 50 and 25 of the dataset respectively. The ablations were performed with Llama-1b trained on GSM8K-Aug dataset and reusing the hyperparameters from Table 6.

# I  INFLUENCE OF GROUND TRUTH TRACES

To assess the effect of ground-truth CoT traces on latent reasoning performance, we used questions from the `GSM8k-Aug-NL` dataset and generated new CoT traces using Qwen3-32B Yang et al. (2025) and Ministral-8B (Mistral AI team, 2024).

For Qwen3-32B, we provided two random examples from the GSM8k training set as few-shot prompts for the model to mimic the style of the traces and disabled the thinking mode. The data was cleaned by removing entries without answers, resulting in 1,165 fewer examples. Qwen3 is known to be verbose, and even with thinking mode disabled, we observed that the generated traces were longer than those produced by GPT-4. The average CoT length increased from 55.0 tokens in the original dataset to 73.5 tokens in the Qwen3-generated version, making it a more challenging task for the latent reasoning model.

In the case of Ministral-8B, we provided three random examples from the GSM8k training set for each generation and generated traces for five questions within one generation. The model was given the correct answer (lacking the ground truth trace) within the prompt. We only preserved generations for which the result and the result of the final equation agreed with the GPT4-generated ground truth, carrying out the procedure up to five times if needed (due to relatively high error rates in this less capable model). If, after five attempts, the generated trace still yielded a different answer than the ground truth or finished in an incorrect equation, we preserved the original, GPT4-generated trace. Furthermore, we removed duplicate (question, answer) pairs from the dataset, preserving only one datapoint for each such pair. The resulting dataset contains 335,126 new (Ministral-generated) datapoints and 50,412 traces from the original dataset. The average CoT length in this dataset version is 44.8 tokens.

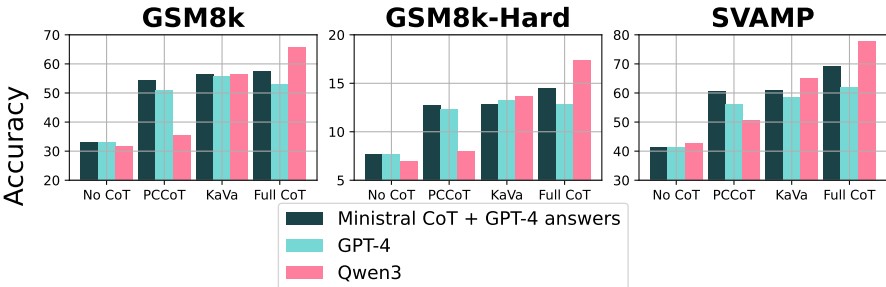

Figure 18: Test accuracies when trained on the original GSM8k-Aug dataset (traces generated by GPT-4) compared to the traces we produces with Qwen3-32B and Ministral-8B model.

To ensure a fair comparison, we use the same hyperparameters as for the original dataset and train KaVa, along with the Full CoT, No CoT, and PCCoT baselines. Test accuracies are reported in Figure 18. We observe that both new datasets generally yield better performance for Full CoT and KaVa. Ministral-generated data differs from the original dataset (GPT-4) only in the reasoning traces, making the No CoT results identical for the two. We hypothesize that shorter traces make the student task slightly easier. For Qwen3, we observe a slight drop in No CoT performance, which suggests that the generated answers may be less accurate than those of GPT-4. Meanwhile, Full CoT performance improves, which may be partially attributed to longer reasoning traces. With this new dataset, KaVa performs slightly better than on GPT-4–generated data across all benchmarks. Notably, unlike Full CoT, we did not adjust the number of latent tokens, thereby maintaining the same level of efficiency.

