# OpenReview forum: "KaVa: Latent Reasoning via Compressed KV-Cache Distillation"
_ICLR.cc/2026/Conference — ICLR 2026 Poster_

### Official Review · Reviewer_JT3U · 2025-10-16

**Soundness:** 2
**Presentation:** 2
**Contribution:** 2
**Rating:** 4
**Confidence:** 4

**Summary:**

This paper introduces KaVa, a method to distill from reasoning chains from large models into smaller ones.
This method can be seen as similar to CODI (published Feb 2025, https://arxiv.org/abs/2502.21074).
But extends CODI by attempting to supervise over the whole chain of reasoning instead of just the ending.
This introduces an issue that the KV Caches for the teacher and student in the distillation are of different sizes.
The authors overcome this by using a KV cache eviction algorithm which is fairly optimal as it can rely on the whole trace (i.e. non-causal) to do eviction as it is not required for inference time.

The authors rely heavily on results from prior work when presenting their evaluations, copying accuracy values.
To validate their method the authors train on two training sets for GSM8K and see some improvement over CODI in most cases, but do achieve a good increase in speed at inference.
The authors ablate some design decisions such as their loss terms and removing the last step of the trace as seen in prior work.

**Strengths:**

- Eviction methods working in KV distillation at all is surprising
- Good speed improvement over CODI, with little/no accuracy compromise.
- Detailed explanation of method.

**Weaknesses:**

- The authors claim their method is novel as it uses eviction methods to reduce the size of the KV cache, unlike KV-Distill (March 2025, https://arxiv.org/pdf/2503.10337). However, the authors do not validate that this is in fact better.
    - This is a key contribution (1 of the 3 listed in section 1), hence I think it should be _a lot_ more rigorously justified.
   - Overall, I think this paper is a little to focused on beating a fixed set of baselines from prior work rather than exploring and rigourously testing their own, new, method.
- There is overlooked prior work on efficiency e.g. https://arxiv.org/pdf/2505.18962 (May 2025)
- Tables 1 and 2 contain a lot of copied values from prior work making it difficult to be sure that the evaluations and training methods are exactly the same. Moreover, there are many gaps in both tables and not all rows from table 1 are included in table 2.
- Limited to gsm8k in both training and testing.
- The authors discuss this method reducing the interpretability of reasoning traces but do not discuss the ethical impacts of this.
- Section 5.2 discusses results entirely in the appendix, slightly against the page limitation rules.

Minor:
- Footnotes for page 5 appear on page 6
- Incorrect bolding in Table 1, 72.4 is bolded and 72.7 is underlined.

**Questions:**

1. Why is an eviction strategy better than KV-Distill empirically?
2. How sure can we be the values copied from prior work in Tables 1 and 2 are identical to the set up used in this paper?

---

> ### Author Response · Authors · 2025-11-21
> **Response to the Reviewer JT3U (1/2)**
>
> We thank the reviewer for highlighting the strengths of our paper, such as the novelty, improvement over CODI and the quality of the explanation.
> We would like to address the questions and weaknesses raised by the reviewer:
>
> * **Q1 \& W1**:
> We believe there is a misunderstanding of our contributions.
> A novel eviction technique is not a contribution of our work.
> KaVa relies on an existing KV-cache compression method. Specifically we use R-KV (NeurIPS 2025) as a strong, training-free eviction approach. The novelty of KaVA (as also pointed out by the reviewers HKZP and XjJK) is that we can distill compressed cache to the latent reasoning student model and generate this compressed KV cache directly at inference. We have updated the contributions section of the paper to make this more clear.
> We rigorously test the performance of the model trained with this distillation method and we provide extensive set of ablation studies. E.g. Figure 6 shows an ablation in which we quantify the impact of changing the compression method on the performance of our technique (we compare R-KV with simpler cosine-based and attention-based eviction strategies as well as with baseline cropping strategy). We are happy to add other specific ablations that the reviewer may find missing.
> We higlight an important distinction between our work and the cited KV-Distill. Our method results in a model which can produce short CoT's (Chains-of-Thought) directly, while the goal of KV-Distill (as well as other compression methods) is to compress a CoT once it has been generated. In particular, we only employ a compression method during the training phase while the inference remains the same as in [3], whereas KV-Distill relies both on specific training and on a specific inference procedure. We have added a clarification on the inference-time distinction of our approach in the related work section.
>
>
>
> * **W2**: Thank you for bringing this concurrent work (accepted for publication at NeurIPS in September 2025) to our attention. We have updated the related work section of our paper accordingly.
> While similarities do exist between [1] and KaVa, the main contributions of the two approaches are largely orthogonal. System1.5 [1] relies on a two-stage pipeline: first, a student model is aligned with a teacher model, and, subsequently a router is learned. While KaVa also relies on a teacher-student setup (where both are the same model but operate in different modes: the teacher reasoning in hard tokens and the student employing latent reasoning), it is an end-to-end approach and obviates the complexity of a multi-stage method. Furthermore, we hypothesize the approach introduced by [1] might benefit from the the kv-matching objective introduced in KaVa in the distillation stage.
> We note that our results are based on larger and more modern models (Llama-3-1B/-3B \& Qwen-2.5-0.5B vs GPT2-125M \& Llama-3-1B in [1]) and include experiments on the more life-like, GSM8K-Aug-NL dataset ([1] utilize only the GSM8K-Aug where the ground-truth traces consist solely of mathematical equations and not free-form text).
> The lack of available source code for [1] makes a comparison with the method infeasible during the rebuttal period.
>
>
> * **Q2 \& W3**:
> We run all the experiments for Qwen-0.5b and Llama-3b ourselves and report all the benchmarks. We only copy the numbers for the Llama-1b experiments.  Some previous works do not report efficiency, resulting in missing lines for those models in Table 2. We do report efficiency for all the experiments that we run ourselves.
> To ensure comparability we largely use the same hyperparameters as reported in in [2,3]: base model, training dataset, batch size, number of epochs, weight decay, LoRA setup, teacher and student loss coefficient, number of latent thoughts and iterations which are provided in [2, 3]. We only vary learning rate and KaVa related parameters.

---

> ### Author Response · Authors · 2025-11-21
> **Response to the Reviewer JT3U (2/2)**
>
> * **W4**: In our experiments we train KaVa on the two versions of the gsm8k dataset and test on three benchmarks: gsm8k, gsm8k-hard and SVAMP. This allows us to compare to related works[2, 3].
> We have made an effort to expand the empirical evaluation by adding one more training dataset (MetaMathQA) and three new evaluation benchmarks (MATH500, multiArith and DeepMind-Mathematics). Consistent with previous results, KaVa demonstrates a superior accuracy–efficiency trade-off compared to prior methods (Fig. 4 and Fig. 16). Additionaly, we observed that KaVa was consistently stable compared to CODI and PCCoT, which had significant training instabilities. Please, see the updated version of the manuscript (Section 4 and Appendix G).
>
> * **W5**: We share the concerns of some portion of the community about the ethical implications of the research on latent reasoning and have decided to update the ethics statement (see the updated version of the manuscript).
>
> * **W6**:  Due to the increased page limit of the rebuttal / camera-ready version, we have been able to move the relevant plot to the main text. Please refer to the updated version of the manuscript.
>
> * **Minor weaknesses (W7 \& W8)**: Thank you for notifying us. We have uploaded a revised version of the manuscript, addressing these oversights.
>
> * **Q1 \& Q2**: We refer the reviewer to the responses to W1 and W3, which answer the raised questions.
>
> Please, let us know if we were able to adress all of your concerns. We are happy to give any further clarification.
>
> References:
>
> - [1] Wang, Xiaoqiang, et al., "System-1.5 Reasoning: Traversal in Language and Latent Spaces with Dynamic Shortcuts"; NeurIPS 2025
>
> - [2] Shen, Zhenyi, et al., "CODI: Compressing Chain-of-Thought into Continuous Space via Self-Distillation'; EMNLP 2025
>
> - [3] Wu, Haoyi, et al., "Parallel Continuous Chain-of-Thought with Jacobi Iteration"; EMNLP 2025

---

> ### Comment · Reviewer_JT3U · 2025-11-23
> **Rebuttal Response**
>
> Weaknesses 1/4/5/6/7/8: Thank you for your informative responses.
>
> > Weakness 2
>
> You refer to R-KV (arxiv: 30 May 2025, NeurIPS 2025) as the basis of your work and system-1.5 (arxiv: 25 May 2025, NeurIPS 2025) as concurrent. This is feels odd.
>
> Your reference on line 231 is also inconsistent with other references to R-KV.
>
> > Weakness 3
>
> I agree it is rigorous to exclude these values if they are not in the same experimental setting. However, this does leave obvious stones unturned in the analysis of KaVa. How difficult would it be to add these values?

---

> > ### Author Response · Authors · 2025-11-26
> > **Response to the Reviewer JT3U**
> >
> > We thank the reviewer for acknowledging our rebuttal. We address the new questions below.
> >
> > **W2**: Thank you for spotting the incorrect reference in line 231 - we have uploaded a fixed version of the manuscript.
> >
> > We agree that both papers (R-KV and System-1.5) were published on arXiv within the same week. However, we happened to not be aware of System-1.5 at the time. We referred to it as concurrent based on the ICLR FAQ definition (though we do not use the term “concurrent” in the manuscript):
> >
> >     Q: Are authors expected to cite and compare with very recent work? What about non peer-reviewed (e.g., ArXiv) papers?
> >     A: We consider papers contemporaneous if they are published within the last two months. That means, since our full paper deadline is September 24, if a paper was published (i.e., at a peer-reviewed venue) on or after July 24, 2025, authors are not required to compare their own work to that paper. Note that arXiv is not considered a peer-reviewed venue. As such, authors are not required to compare to papers solely on arXiv: they may be excused for not knowing about papers not published in peer-reviewed conference proceedings or journals, which includes papers exclusively available on arXiv.
> >
> > For visibility, the NeurIPS 2025 notification deadline was September 18. As per reviewer's request we added this work in the related work section of our revised manuscript. Please let us know if you have any further recommendations.
> >
> > **W3**: We would like to emphasize that we did not exclude any numbers. On the contrary, we made every effort to present all available results from prior work. Blank spaces were only left where results were not reported (for example, CODI only reports the number of tokens in the GSM8K dataset). Following the reviewers’ suggestion, we have re-run the experiments ourselves to report the missing numbers.
> >
> > We can split this concern in two parts:
> >
> > - *W3.1: Missing numbers and  for LLaMA-1B:*
> > We have updated Tables 1 and 2 to include results obtained by re-running CODI, PCCoT, Full CoT, and No CoT using our pipeline, while keeping the originally reported results in parentheses. We observe that our results are largely consistent with prior work.
> > Due to the completely different training pipeline (curriculum learning), we did not re-run iCoT and Coconut.
> >
> > - *W3.2: Not all rows from Table 1 are included in Table 2:*
> > The methods we compare with can be grouped together, based on how they behave in inference: CoT - group 1,  No-CoT / iCoT - group 2, CODI / Coconut - group 3, PCCoT / KaVa - group 4. Assuming perfect token-level response accuracy, the number of forward passes for all methods within a group would be identical. In practice, slight differences may occur due to format inconsistencies or variations in answer length. As we expect these differences to be minimal, we opted for brevity by including measurements for only one representative method per group. We made the rationale clearer in an updated version of the manuscript. We are also open to incorporating any other improvements.

---

> > > ### Comment · Reviewer_JT3U · 2025-11-27
> > >
> > > Thank you for the response.
> > >
> > > I agree the word "exclude" has the wrong connotation in this case, and agree the authors took the best course of action by leaving blank spaces. It's great to see most of these filled in.
> > >
> > > I agree with MUtU that the open source experiments are interesting and could really improve the value of the paper, as we will understand more about the role of the teacher.
> > > Overall, post rebuttal I think my concerns are addressed and will increase my score.

---

> > > > ### Author Response · Authors · 2025-11-28
> > > >
> > > > Thank you for your reply and raising your score to 8. We are glad that our new addition of the MetaMathQA training dataset, the new evaluation benchmarks (MATH500, MultiArith, and DeepMind-Mathematics), and the new experimental results using LLaMA-1B, effectively addressed your concerns and helped strengthen the paper. In addition, we are currently working on the experiments requested by the reviewer MUtU.

---

### Official Review · Reviewer_MUtU · 2025-10-23

**Soundness:** 3
**Presentation:** 3
**Contribution:** 3
**Rating:** 4
**Confidence:** 3

**Summary:**

This paper introduces KAVA, the first framework that distills knowledge directly from the compressed KV-cache of the teacher into a latent-reasoning student via self-distillation. This process is conducted in a step-wise fashion, removing the eviting KVs that do not are less important.

**Strengths:**

- interesting idea to distill from the KV-cache
- Overall, good comparisons with other baselines

**Weaknesses:**

- The biggest weakness of this paper is that the performance of the teacher caps any distillation process.
- Lack of diversity in downstream tasks
- Writing
   - Move Figure 1 earlier in the paper (i.e pg 1-2)
   - The main benefit of this method seems to be the reduction of the inference cost. However, this result/point is presented over 2 pages instead of just having a single graph/table showing the Pareto frontier curve

**Questions:**

- Why is there not a complete comparison between all the methods and models? Is this from a resource limitation? I would suggest moving those comparisons to another table.
- How does the quantity or quality of the data affect the distillation? How can this method move beyond the limitations of current distillation?
- How does the distillation process compare when distilled from its own CoT or a stronger model?

---

> ### Author Response · Authors · 2025-11-21
>
> We thank the reviewer for the thoughtful review. We are glad that the reviewer found our method to be interesting and is happy with the baselines we provide. Below we address questions and concerns raised by the reviewer.
>
> * **W1 \& Q2**.  We agree that the student performance in the distillation process is limited to some extent by the teacher performance. This is, however, a general limitation of any distillation method.
> However, this limitation is not strict, since distilling into a shorter reasoning trace can have a regularizing effect leading to a better generalization. For example, we observe that KaVa performs slightly better than Full CoT (which is a proxy for teacher performance) on GSM8k in Llama-1b setting (see Table 1 in the paper).
> Furthermore, it is unclear what is the cap on the teacher's performance: the traces come from a better model (GPT4), and perhaps with enough training data both the teacher and the student could achieve better (and similar) performance.
> Also, we note that due to our method's good accuracy in comparison with prior work, it can be considered a step towards closing the gap between the student and the teacher.
>
>
> * **W2**. To address this weakness, we extend the experimental evaluation to a more complex training dataset (MetaMathQA) and add three new evaluation benchmarks: multiArith, Math500, and DM-Math. Please see the general response and updated manuscript for more details.Consistent with previous results, KaVa demonstrates a superior accuracy–efficiency trade-off compared to prior methods (Fig. 4 and Fig. 16). Additionaly, we observed that KaVa was consistently stable compared to CODI and PCCoT, which had significant training instabilities. Please, see the updated version of the manuscript (Section 4 and Appendix G).
>
> * **W3.1 (Writing)**. Thank you for this suggestion, we have moved Figure 1 to page 2.
>
> * **W3.2 (Writing)**. We agree that a Pareto frontier plot is very informative. Using the new MetaMathQA dataset, we vary the number of CoT tokens replaced by latent thinking, which allows us to draw a Pareto curve. This demonstrates that KaVa requires far fewer tokens to generate answers, improving efficiency while still outperforming no-CoT, CODI, and PCCoT (Figure 4 in the updated manuscript).
> Additionally, we include an accuracy–efficiency analysis of the previous results in Appendix F and reference it in the main text (Section 4.2).
> We prefer to retain the two-table format in the main body, as it enables comparison with methods that do not report efficiency. For convenience, we have moved Table 2 to appear on the same page as Table 1.
>
>
> * **Q1**.  We focused on training all the baselines for Llama-3b and Qwen-0.5b ourselves, since those are not reported in the original papers. For Llama-1b, we copy the numbers from the corresponding papers. Unfortunately, they do not report some of the benchmark results for the GSM8k-Aug-NL dataset, resulting in the gaps in the tables. We are afraid a separate table for Llama-1b result will take more space.
>
> * **Q2**. Both data quality and quantity are critical for effective distillation. To assess the impact of **quantity** empirically, we run an experiment where we use smaller subsets of the GSM8k-Aug dataset for training. We observe that a large dataset is crucial for good performance of our method. For specifics, please refer to the App. H in the updated manuscript.
> When it comes to **quality**, we can compare the results on GSM8k-Aug (a ‘cleaned’ version retaining only essential equations) with GSM8k-Aug-NL (non-cleaned). Models trained on the NL version of the dataset always produce lower accuracy than the one trained on the simple one. However, KaVa demonstrates a smaller drop in performance compared to the baselines.
>
>
> * **Q3**. In our method, we use ground-truth CoT to produce the KV-cache and apply compression. These ground truth CoT's were generated by GPT-4 in the case of the GSM8k-Aug dataset and by GPT-3.5-Turbo in the case of the MetaMathQA dataset.
> To distill from the model's own CoT we would need to autoregressively generate it during training, which is unfortunately prohibitively expensive and will probably slow down the convergence, since the teacher is not able to produce good enough CoT traces in the beginning of training.
>
>
> Please, let us know if we were able to address all of your concerns. We are happy to give any further clarification.

---

> ### Comment · Reviewer_MUtU · 2025-11-25
>
> Thank you for your rebuttal. W1 and Q2 make sense, and I appreciate the writing updates. The Pareto frontier plot looks great. I am more at a five based on the current discussion (note five is actually not a score so I will keep my current rating at this time).
>
> > W2 lack of diversity in downstream tasks
>
> Why were they tasks chosen? Why not non-Math tasks?
>
> > Q2: Data Quality
>
> App H is great, but I think it would be even better if there were a baseline (or even a couple) on this plot. Note that one baseline is fine as time and resources are limited in the rebuttal period, but for a future version, multiple baselines would be ideal.
>
> > Q3: teacher model
>
> I don't think this answers my question. I really want to see a plot here to quantify some difference. If speed is a concern, one could do a comparison between GPT-5 and GPT-4, including some sort of baseline if applicable would be informative here as well.

---

> > ### Comment · Reviewer_MUtU · 2025-11-25
> >
> > Note, I just remembered five is not a valid option, so  I will remain at a score of four. However, I really am at five. I edited the response above. However, email updates do not record edits. I am commenting on this as well, so the authors are correctly notified.

---

> > > ### Author Response · Authors · 2025-11-26
> > > **Response to the Reviewer MUtU**
> > >
> > > We thank the reviewer for acknowledging our rebuttal. We address the new questions below. In addition, we have updated the manuscript with the additional experiments requested by reviewer JT3U. Specifically, we re-ran the CODI and PCCoT baselines to address the gaps in Tables 1 and 2, which appeared due to missing baselines reported in prior work. This also addresses Q1 from your original review.
> > >
> > > > Why were they tasks chosen? Why not non-Math tasks?
> > >
> > > We focused on mathematical reasoning (GSM8K, MetaMath, etc.) because this is the standard evaluation protocol established by prior latent reasoning works [1, 2, 3, 4, 5]. Since the training data consists exclusively of math problems, we do not expect the reasoning to generalize to non-math-related questions.
> > >
> > >
> > > > Q2: Data Quality
> > >
> > > Thank you for the suggestion. We are currently running the *PCCoT baseline* for the data scaling experiment (Appendix H) and will add this to the plot in the final version. We would like to clarify whether this is the baseline the reviewer is referring to. If we know the exact expectations, it will be easier for us to run the correct experiments and provide the results on time.
> > >
> > > > Q3: teacher model
> > >
> > > **Clarification on Scope:**
> > > Our paper investigates *how efficiently* a student can internalize a given teacher's reasoning. The "Full CoT" results in our tables already serve as the proxy for the "Teacher" performance. We show that KaVa closes the gap to this teacher more effectively than baselines. Using a stronger teacher (like GPT-5) would likely shift the performance ceiling (Full CoT) upwards for all methods, but it would not fundamentally change the relative efficiency analysis of the *distillation mechanism* itself, which is our core contribution.
> > >
> > > We understand the reviewer’s interest in quantifying how a different teacher strength affects distillation performance. However, we would like to clarify two key practical constraints that prevent us from including the specific plot requested:
> > >
> > > 1.  **GPT-5 Access:** GPT-5 was released only a month prior to the submission deadline. We do not have API access to this model. Consequently, generating a new dataset with GPT-5 is not possible for us. Furthermore, our current datasets (generated by GPT-4 and GPT-3.5) are standard open-source benchmarks; creating a custom dataset would decouple our results from the literature, making fair comparison impossible.
> > > 2.  **Self-Distillation Cost:** Regarding distilling from the model's "own CoT": as mentioned, this requires the model to autoregressively generate full reasoning traces *online* during training to serve as the teacher signal. This increases the training compute budget by a factor of $N$ (the length of the trace) compared to our method, which uses pre-computed ground-truth traces. This would be prohibitively expensive.
> > >
> > >
> > > We hope this explanation clarifies why we cannot provide the requested plot involving GPT-5 or online self-distillation. If the reviewer have a specific and feasible experiment in mind, we are happy to add it.
> > >
> > >
> > >
> > > ### References
> > > [1] Deng, Yuntian, Yejin Choi, and Stuart M. Shieber. "From Explicit CoT to Implicit CoT: Learning to Internalize CoT Step by Step." CoRR 2024.
> > >
> > > [2] Hao, S., Sukhbaatar, S., Su, D., Li, X., Hu, Z., Weston, J., & Tian, Y.. Training large language models to reason in a continuous latent space. arXiv preprint; 2024.
> > >
> > > [3] Shen, Zhenyi, et al., "CODI: Compressing Chain-of-Thought into Continuous Space via Self-Distillation'; EMNLP 2025
> > >
> > > [4] Wu, Haoyi, et al., "Parallel Continuous Chain-of-Thought with Jacobi Iteration"; EMNLP 2025
> > >
> > > [5] Su, D., Zhu, H., Xu, Y., Jiao, J., Tian, Y., & Zheng, Q. Token Assorted: Mixing Latent and Text Tokens for Improved Language Model Reasoning. ICML 2025

---

> > > > ### Comment · Reviewer_MUtU · 2025-11-26
> > > >
> > > > Thank you for your response!
> > > >
> > > > > Q2
> > > >
> > > > PCCoT works as a baseline, as KAVA is an extension of PCCoT; just having some comparison point is useful for the reader. Either CODI or PCCoT would have been fine with me.
> > > >
> > > > > Q3
> > > >
> > > > Thank you for the clarification. It might be possible to use open-source models to help understand how different teachers' CoT might affect the model. I think for distillation papers is particularly important to understand how the teacher's strength affects the method. I do think GPT-3.5 is an incredibly weak model by today's standards. However, some open-source models are available that might be useful here (Qwen3, DeepSeek, Llama3.1, or Llama 4). TogetherAI's API might be useful for this experiment. My goal of this ask is to gain a comprehensive understanding of how the teacher model affects KAVA.
> > > >
> > > > From my understanding, KAVA requires just the CoT from the model and then uses the self-distillation over the KV-Cache. I hypothesize that the CoTs from a stronger model are a much stronger signal on the kv-cache distillation. So, I do believe these experiments are possible. Now, with not much time left in the rebuttal period, I understand a "comprehensive understanding" might not be feasible. So, I'm willing to see two different experiments with two different models that are clearly both performant on the task, but one is on paper much better than the other on other tasks. I would like to see in the camera-ready a more comprehensive set of these experiments.

---

> > > > > ### Author Response · Authors · 2025-12-03
> > > > > **Response to the Reviewer MUtU**
> > > > >
> > > > > Thank you for your clarifications. We run the requested experiments and updated the manuscript. We summaruze the changes and new results below.
> > > > >
> > > > > > Q2
> > > > >
> > > > > As discussed, we have added the results for PCCoT trained on a smaller dataset (Appendix H, Figure 17, bottom). These results suggest that PCCoT performance declines more sharply when the amount of training data is reduced, although it benefits slightly more from extended training. However, even after 40 epochs, PCCoT accuracy remains behind KaVa trained for only 10 epochs.
> > > > >
> > > > > > Q3
> > > > >
> > > > > We thank the reviewer for the clarification and did our best to conduct the requested experiment with the available resources. We used the Qwen3-32B model to generate new ground-truth traces, assuming it is sufficiently capable given its 93% accuracy on the GSM8k dataset (as reported in Table 4 of [1]).
> > > > > We use 385,620 questions form the GSM8k-Aug dataset, and the whole generation took approximately two days.We subsequently removed 1,165 questions for which no answer was generated.
> > > > > As a second data point, we used the Ministral-8B model. Since this is a less capable model, we generated only reasoning traces and used the correct answers from the GPT-4–generated dataset. The full generation procedure is described in Appendix I of the updated manuscript.
> > > > >
> > > > >
> > > > > To ensure a fair comparison, we applied the same hyperparameters as for the original dataset and trained KaVa, along with the Full CoT, No CoT, and PCCoT baselines. The results have been added to Appendix I in the updated manuscript. We obtained better Full CoT and slightly better KaVa results. For the Qwen3-generated dataset, we observed slightly lower No CoT accuracy and a significant drop in PCCoT performance.
> > > > >
> > > > >  [1] Yang, A., Li, A., Yang, B., Zhang, B., Hui, B., Zheng, B., ... & Qiu, Z. (2025). Qwen3 technical report.

---

### Official Review · Reviewer_XjJK · 2025-11-01

**Soundness:** 3
**Presentation:** 3
**Contribution:** 2
**Rating:** 6
**Confidence:** 3

**Summary:**

This paper introduces KAVA, a framework that distills knowledge from a teacher model’s compressed KV-cache into a latent reasoning student through self-distillation. By aligning continuous latent representations, KAVA enables efficient and supervised latent reasoning without explicit chain-of-thoughts. Experiments show that it outperforms strong baselines, maintaining high accuracy and scalability while significantly improving inference efficiency.

**Strengths:**

1. The paper presents a novel and timely attempt to bridge explicit chain-of-thought reasoning and latent reasoning through compressed KV-cache distillation, offering a promising direction for improving efficiency without sacrificing reasoning quality.

2. The proposed KAVA framework is well-motivated and technically sound, combining self-distillation with KV-cache compression to provide an effective supervision signal for latent reasoning models.

3. The experiments are comprehensive and show consistent improvements over strong baselines such as CODI and PCCoT, demonstrating both better reasoning performance and significant inference-time efficiency gains.

**Weaknesses:**

1. The paper demonstrates the potential of applying KV-cache compression to latent reasoning, but it does not clearly explain how its KV-cache design differs from or improves upon existing approaches.

2. All experiments are conducted on relatively small models (0.5B–3B), without evaluating performance on larger-scale models. Is this due to computational limitations, algorithmic instability, or other design considerations?

3. While the paper claims that the compressed KV-cache serves as a “rich supervision signal,” it lacks theoretical or information-theoretic analysis to justify why KV representations would preserve reasoning structure better than hidden states.

**Questions:**

See Weaknesses.

---

> ### Author Response · Authors · 2025-11-21
>
> We thank the reviewer for the thoughtful feedback. We are pleased that the reviewer found our method to be novel, well-motivated, and supported by a comprehensive experimental evaluation. Below, we address the questions and concerns raised:
>
> * **W1**: We would like to clarify that we are not proposing a new KV-cache compression or design method. Instead, we leverage existing compression techniques to produce a supervision signal for the latent reasoning model. This enables us to supervise all intermediate steps of latent reasoning rather than only the activations of the final token. We have updated the second contribution point in the paper to make this clearer.
> Please, let us know if this addresses your concern, we are happy to give further clarifications.
>
> * **W2**: The largest model used in prior works (CODI, PCCoT) is LLaMA-1B. Adding larger models requires training not only for our method but also all baselines. For example, for LLaMA-3B, we separately tuned the learning rate and trained all methods (No CoT, Full CoT, CODI, and PCCoT) on two training datasets. Training all methods on LLaMA-8B would unfortunately exceed our computational budget. We did not observe any instability when scaling from LLaMA-1B to LLaMA-3B for our method, thus we do not expect any additional issues when scaling it further.
>
> * **W3**: We assume that our method provides a better supervision signal primarily because it is more dense: we supervise all intermediate steps of latent reasoning, whereas previous methods supervise only the final step.
> Consider the following toy example: let $q$ be a question token with corresponding KV-cache $KV_q$, $d$ is a distillation token (1 token before the answer), and $a$ the answer token. Assume we have three latent iterations: $z_1,  z_2, z_3$.  Let $f^{LLM}$ denote a forward pass of the LLM.  Then, the full forward pass of the student consists of four autoregressive steps:
> $$ \text{Step }1: \, z_2; KV_{z_1} = f^{LLM}(KV_q, z_1) $$
> $$ \text{Step }2: \, z_3; KV_{z_2} = f^{LLM}(KV_q, KV_{z_1}, z_2) $$
> $$ \text{Step }3: \, \hat{d}; KV_{z_3} = f^{LLM}(KV_q, KV_{z_1, z_2}, z_3) $$
> $$ \text{Step }4: \, \hat{a}; KV_{d} = f^{LLM}(KV_q, KV_{z_1, z_2, z_3}, d) $$
> CODI’s distillation loss only uses activations from the last step, as it is the only step with full token correspondence between teacher and student. Therefore, steps 1–3 lack intermediate supervision. In contrast, our method provides supervision signals for $KV_{z_1}, KV_{z_2}, KV_{z_3}$, matching them to the compressed KV-cache of the teacher.
> Theoretically, matching the final token activation could suffice to produce the correct answer. Empirically, we observe that adding our distillation loss consistently improves model performance.
>
> Please, let us know if we were able to address all of your concerns. We are happy to give any further clarification.

---

### Official Review · Reviewer_HKZP · 2025-11-01

**Soundness:** 3
**Presentation:** 4
**Contribution:** 3
**Rating:** 8
**Confidence:** 4

**Summary:**

The paper proposes a novel way to enable latent reasoning capabilities for thinking models. LLMs with explicit CoT are known to excel today at highly challenging multi-step reasoning problems. However, the explicit CoT paradigm is very expensive and requires a lot of computational and memory overhead during inference. To reduce this, reasoning in the latent space has been suggested as an alternative. However, the methods suggested so far for latent reasoning, suffer quality losses compared to explicit CoT. The paper posits that a main reason for this is we cannot provide supervision in the latent space. To remedy this, the authors propose KAVA: a method that takes in a teacher model with explicit CoT, compresses its KV cache, then distills the compressed KV cache down to a student model (self-distillation - student model is same size as the teacher model) to enable latent reasoning abilities.

KAVA relies on some recent studies which show that KV caches underlying CoT are highly redundant and can be compressed heavily along the sequence length dimension. This motivates the possibility of providing supervision for latent reasoning via a compressed KV cache. The training for KAVA proceeds as follows:
1. The tokens are split into 3 parts: question Q, reasoning trace C, answer A.
2. We then apply a redundancy-aware KV-cache compression scheme to the teacher cache of the reasoning trace. This works by computing an importance and a redundancy score for each token and choosing the top M tokens to match the student’s latent reasoning length. The importance score of a token is simply its attention weight. The redundancy is computed as the average pairwise cosine similarity among all keys and normalized by a softmax.
3. Then we perform distillation of these compressed KV caches to the student using an L_p loss on the embeddings. The p is chosen by hyper parameter sweep and ends up being either 1 or 2 depending on the specific dataset and model. The student’s latent embeddings are generated in parallel using Jacobi decoding for 3 iterations.


The authors apply this method to train fine-tuned latent reasoning student models from Llama 3.2, 1B and 3B pretrained models and Qwen 2.5 0.5 B pretrained model.
They use LoRA fine-tuning with rank 128. They fine-tune the models on GSM8k-AUG and GSM8k-AUG-NL datasets which have ~400k examples (augmented versions of GSM8k) with CoT generated using GPT-4. GSM8k-AUG-NL is in natural language while GSM8k-AUG removes the natural language and only keeps equations.
Then they evaluate on the original GSM8k, GSM8k-Hard and SVAMP datasets. KAVA reduced the thinking trace size by around 80-90% while outperforming other latent reasoning methods. For instance, the Llama 1B model with full CoT gets 61% on GSM8k, and 30.9% with no CoT. KAVA gets 56.5% beating the next best latent reasoning method by 3%.

The authors ablate for different choices of the KV eviction loss, the L_p loss and other modeling choices.
They also perform an interpretability analysis of the latent reasoning traces but it’s unclear if the presented results offer statistically convincing arguments for the inferences made. The paper concludes with a nice visualization into the similarity of the teacher and student KV caches after distillation. They observe a higher similarity between a latent representation at position n with explicit CoT representations at positions m for n < m which is the behavior we want.

**Strengths:**

- The paper tackles an important and highly relevant problem for today’s LLM landscape. The proposed solution is novel and practical.
- The paper performs a good number of ablations for different hyper parameter choices. While it would have been nice to see experiments on more datasets than GSM8k, the choice seems guided by a number of past works in this space choosing to focus on GSM8k for models in the 1-2B size range.

**Weaknesses:**

- The main weakness is the limiting choice of the dataset. Ablations on larger models on more challenging datasets would have given stronger evidence to the efficacy and scalability of the proposed solution. The proposed method is interesting and promising but whether it works across datasets and scale is not fully established.

- The focus on fine-tuning and evaluating on a specific dataset or domain of problems raises the question on whether the proposed method hurts other abilities of the LLM. A more broader downstream evaluation across a suite of benchmarks would present evidence that we aren’t simply trading off one ability for another within the model. This perhaps would require moving away from tasks-specific LoRA fine-tuning.

**Questions:**

- It would also be interesting to see what would happen with full-fine tuning instead of just LoRA fine-tuning. Perhaps the gap to full CoT can be bridged in this case.

---

> ### Author Response · Authors · 2025-11-21
>
> We thank the reviewer for the thoughtful review. We are glad that the reviewer found our method to be novel and practical and appreciated the diverse ablation studies that we performed. Below we address questions and concerns raised by the reviewer.
>
> * **W1**: To address this weakness, we extend our experimental section by adding one more training dataset: MetaMathQA, which includes (1) longer reasoning traces and (2) harder tasks (including problems from the training split of MATH). Consistent with previous results, KaVa demonstrates a superior accuracy–efficiency trade-off compared to prior methods (Fig. 4 and Fig. 16). Additionaly, we observed that KaVa was consistently stable compared to CODI and PCCoT, which had significant training instabilities. Please, see the updated version of the manuscript (Section 4 and Appendix G).
>
>
> * **W2**: With a new training dataset we were able to extend the list of evaluation benchmarks we use. We added three new datasets: multiArith, Math500 and DM-Math (Fig. 4 and Fig. 16). Please, see the general response and updated manuscript for more details.
>
>
> * **Q1**: Thank you for this question. We used LoRA to have a fair comparison with all the previous methods. Furthermore [1] (App. C.2) suggests that using LoRA might be crucial to avoid overfitting while still being able to perform a large number of training steps. As proposed by the reviewer we have performed additional experiments replacing LoRA with full-parameter finetuning and noted a significant drop in performance (method: original\_score -> new\_score): CoT: 61.6 -> 37.9, no-CoT: 30.9 -> 4.6, PCCoT: 53.35 -> 3.5, KaVa 56.5 -> 7. We hypothesize that with a more extensive hyperparameter search this performance could be improved (we reuse the hyperparameters from App. C, Table 6), however, we find this infeasible within the short timeframe of the rebuttal period.
> If the reviewer believes the specifics of this ablation would benefit the community, we would be willing to include it in the camera-ready version of the paper.
>
>
> Please, let us know if we were able to adress all of your concerns. We are happy to give any further clarification.
>
> References:
>
> - [1] Wu et al., "Parallel Continuous Chain-of-Thought with Jacobi Iteration"; EMNLP 2025

---

### Author Response · Authors · 2025-11-21
**General Response**

We thank reviewers for their valuable feedback and thorough evaluations which allowed us to strengthen the paper. We upload updated version of the manuscript. Below, we summarize the main points raised and outline the additions and revisions made to the paper.

## Method Novelty and Contribution
We are delighted that reviewers found our method to be novel (HKZP, XjJK), interesting (MUtU), timely (XjJK), and practical (HKZP).

### Improvements Made
We have updated the Contributions and Related Work sections to highlight some distinctions with priors work and our contribution.
We clarify that KV-distillation is not equivalent to KV-cache design or compression (JT3U, XjJK). Our approach leverages existing KV-cache compression techniques to provide an effective and dense supervision signal for latent reasoning models. Unlike prior latent reasoning methods that supervise only the final token activation, our method supervises all intermediate reasoning steps, which we believe is key to improving reasoning quality. Furthermore, unlike KV-compression methods, we compress the KV-cache only during training; at inference, the student generates short CoT traces directly.


## Performance

Reviewer HKZP highlighted the consistent improvement over strong baselines, and both JT3U and HKZP appreciated the efficiency gains of our method. We are pleased that reviewer JT3U was surprised by the performance of our approach, which we take as a positive indication of its impact.

## Experimental Setup

Our experimental evaluation was considered comprehensive (XjJK) and includes strong comparisons with baselines (MUtU) and diverse ablation studies (HKZP). As fairly noted by reviewer HKZP, our setup was primarily motivated by prior work: we use the same training datasets and benchmarks for fair comparison. However, we extend the evaluation to models not previously used, adding Qwen-2.5-0.5B and LLaMA-3-3B.

### Improvements Made

The choice of datasets was mentioned as a weakness (HKZP, JT3U), along with the lack of diversity in downstream tasks (JT3U, MUtU). To address these concerns, we extend our experimental section by adding one more training dataset: MetaMathQA [1], which includes (1) longer reasoning traces and (2) harder tasks (including problems from the training split of MATH [2]). Furthermore, we add three new benchmarks: MATH500 [2], DeepMind Mathematics [3], and MultiArith [4]. Experimental setup and the new results are added to Section 4 and Appendix G of the updates manuscript.
Furthermore, we use these experiments to incorporate the request of reviewer MUtU and present an Accuracy-Effiency Pareto curve (Fig. 4 and Fig.16).
Consistent with previous results, KaVa demonstrates a superior accuracy–efficiency trade-off compared to prior methods.
Additionaly, we observed that KaVa was consistently stable compared to CODI and PCCoT, which had significant training instabilities.


## Other Additions to the Paper
Additional improvements in the revised paper include:

* A new ablation: impact of data scaling, Appendix H (MUtU)

* Extra Pareto-frontier presentation of the previous results, Appendix F (MUtU)

* Added concurrent work (accepted for publication in September 2025) to the related work Section 2 (JT3U)

* Updated the "Ethics and Reproducibility Statement" (JT3U)

* Move Fig. 8 to the main body of the paper (JT3U)

* Move Fig. 1 ealier in the paper (MUtU)

References:

[1] Yu, Longhui, et al. "MetaMath: Bootstrap Your Own Mathematical Questions for Large Language Models." ICLR 2024

[2] Hendrycks, Dan, et al. "Measuring Mathematical Problem Solving With the MATH Dataset."  NeurIPS 2021

[3] Saxton, David, et al. "Analysing Mathematical Reasoning Abilities of Neural Models." ICLR 2019

[4] Roy, Subhro, and Roth, Dan. "Solving general arithmetic word problems." EMNLP 2015

---

> ### Author Response · Authors · 2025-12-03
> **Summary of the Discussion**
>
> We would like to summarize the outcomes of the discussion period and the interactions we had with reviewers while it was still possible. As a result of these exchanges, we made several adjustments beyond the original rebuttal, which are detailed below.
>
> #### Reviewer JT3U
> The reviewer requested Llama-1B results for the baselines under exactly the same settings as our method. To address this, we re-ran PCCoT and CODI (prior works) and reported the results in addition to copying the numbers from the corresponding papers. We have updated Tables 1 and 2 in the manuscript to include these results
>
> This resolved the reviewer’s final concerns, as confirmed in their last response.
>
> #### Reviewer MUtU
> The reviewer was content with the most of the rebuttal. Additional experiments where requested during the discussion period which we added to the paper and that we summarize below.
>
> Q2. Subset Training:
> We were asked to add a baseline to the subset training experiments (Appendix H). We have updated the corresponding appendix to include results for the PCCoT baseline.
>
>
> Q3. Effect of Different Ground-Truth CoT:
> The reviewer requested a quantitive analysis of the effect of ground-truth CoT on the method performance. To address this, we generated a new training dataset using the open-source Qwen3-32B and Ministral-8B models and trained the proposed approach as well as relevant baselines. These new experiments have been added to Appendix I.
>
> #### Reviewers HKZP & XjJK
> Two other reviewers did not respond during the discussion period but had both originally recommended acceptance of the paper.

---

### Meta-Review · Area_Chair_V4Jj · 2026-01-14

**Summary:**

The paper proposes a novel framework, KaVa, which distills knowledge from a teacher's compressed KV-cache into a student model to enable efficient latent reasoning. This approach tackles a key issue in latent reasoning: the lack of a dense supervision signal. The consensus among reviewers is positive, with scores improving during the rebuttal phase.

The reviewers, in general, had positive feedback about the paper. While there were concerns about the empirical analysis, the authors's rebuttal address most of their concerns. I recommend acceptance in the current form.

**Reviewer Concerns:**

Reviewer JT3U raised concern about missing baseline values for LLaMA-1B in tables. Concern that the method wasn't rigorously compared against "KV-Distill" and "System 1.5". The authors re-ran PCCoT and CODI baselines to fill gaps in the tables, ensuring a fair "apples-to-apples" comparison. The authors sufficiently addressed the concerns of the reviewer.

Reviewer MUtU raised concerns about limited experiments and about performance being capped by teacher's performance (which is probably true for any distillation setup). The reviewer also requested a Pareto frontier of Accuracy vs. Efficiency, which the authors added.

**Reviewer Scores:**

All the reviewers have given positive score. Reviewer JT3U increased their score after requested experiments were added.  I think given the additional experiments, Reviewer MUtU would probably raise their score to 6. In such a scenario, the paper has overall positive scores.

---

### Decision · Program_Chairs · 2026-01-26

Accept (Poster)